# Optimal Client Sampling for Federated Learning

**Wenlin Chen**                                                          *wc337@cam.ac.uk*
*Department of Engineering*
*University of Cambridge*
*Cambridge, CB2 1PZ, UK*
*and*
*Department of Empirical Inference*
*Max Planck Institute for Intelligent Systems*
*Tübingen, 72076, Germany*

**Samuel Horváth**                                      *samuel.horvath@mbzuai.ac.ae*
*Department of Machine Learning*
*Mohamed bin Zayed University of Artificial Intelligence*
*Masdar City, Abu Dhabi, UAE*

**Peter Richtárik**                                        *peter.richtarik@kaust.edu.sa*
*Computer, Electrical and Mathematical Science and Engineering Division*
*King Abdullah University of Science and Technology*
*Thuwal, 23955-6900, Saudi Arabia*

**Reviewed on OpenReview:** *https://openreview.net/forum?id=8GvRCWKHIL*

## Abstract

It is well understood that client-master communication can be a primary bottleneck in federated learning (FL). In this work, we address this issue with a novel client subsampling scheme, where we restrict the number of clients allowed to communicate their updates back to the master node. In each communication round, all participating clients compute their updates, but only the ones with "important" updates communicate back to the master. We show that importance can be measured using only the norm of the update and give a formula for optimal client participation. This formula minimizes the distance between the full update, where all clients participate, and our limited update, where the number of participating clients is restricted. In addition, we provide a simple algorithm that approximates the optimal formula for client participation, which allows for secure aggregation and stateless clients, and thus does not compromise client privacy. We show both theoretically and empirically that for Distributed SGD (DSGD) and Federated Averaging (FedAvg), the performance of our approach can be close to full participation and superior to the baseline where participating clients are sampled uniformly. Moreover, our approach is orthogonal to and compatible with existing methods for reducing communication overhead, such as local methods and communication compression methods.

## 1 Introduction

We consider the standard cross-device federated learning (FL) setting (Kairouz et al., 2019), where the objective is of the form

$$\min_{x \in \mathbb{R}^d} \left[ f(x) := \sum_{i=1}^{n} w_i f_i(x) \right], \tag{1}$$

where $x \in \mathbb{R}^d$ represents the parameters of a statistical model we aim to find, $n$ is the total number of clients, each $f_i \colon \mathbb{R}^d \to \mathbb{R}$ is a continuously differentiable local loss function which depends on the data distribution

$\mathcal{D}_i$ owned by client $i$ via $f_i(x) = \mathrm{E}_{\xi \sim \mathcal{D}_i}[f(x, \xi)]$, and $w_i \geq 0$ are client weights such that $\sum_{i=1}^n w_i = 1$. We assume the classical FL setup in which a central master (server) orchestrates the training by securely aggregating updates (Du & Atallah, 2001; Goryczka & Xiong, 2015; Bonawitz et al., 2017; So et al., 2021), i.e., the master only has access to the sum of updates from clients without seeing the raw data.

## 1.1 Motivation: Communication Bottleneck in Federated Learning

It is well understood that communication cost can be a primary bottleneck in cross-device FL, since typical clients are mobile phones or different IoT devices that have limited bandwidth and availability for connection (Van Berkel, 2009; Huang et al., 2013). Indeed, wireless links and other end-user internet connections typically operate at lower rates than intra-datacenter or inter-datacenter links and can be potentially expensive and unreliable. Moreover, the capacity of the aggregating master and other FL system considerations imposes direct or indirect constrains on the number of clients allowed to participate in each communication round. These considerations have led to significant interest in reducing the communication bandwidth of FL systems.

**Local Methods.** One of the most popular strategies is to reduce the frequency of communication and put more emphasis on computation. This is usually achieved by asking the devices to perform multiple local steps before communicating their updates. A prototype method in this category is the Federated Averaging (`FedAvg`) algorithm (McMahan et al., 2017), an adaption of local-update to parallel SGD, where each client runs some number of SGD steps locally before local updates are averaged to form the global update for the global model on the master. The original work was a heuristic, offering no theoretical guarantees, which motivated the community to try to understand the method and various existing and new variants theoretically (Stich, 2019; Lin et al., 2018; Karimireddy et al., 2019; Stich & Karimireddy, 2020; Khaled et al., 2020; Hanzely & Richtárik, 2020).

**Communication Compression Methods.** Another popular approach is to reduce the size of the object (typically gradients) communicated from clients to the master. This approach is referred to as gradient/communication *compression*. In this approach, instead of transmitting the full-dimensional gradient/update vector $g \in \mathbb{R}^d$, one transmits a compressed vector $\mathcal{C}(g)$, where $\mathcal{C} : \mathbb{R}^d \to \mathbb{R}^d$ is a (possibly random) operator chosen such that $\mathcal{C}(g)$ can be represented using fewer bits, for instance by using limited bit representation (quantization) or by enforcing sparsity (sparsification). A particularly popular class of quantization operators is based on random dithering (Goodall, 1951; Roberts, 1962); see Alistarh et al. (2017); Wen et al. (2017); Zhang et al. (2017); Ramezani-Kebrya et al. (2019). A new variant of random dithering developed in Horváth et al. (2019) offers an exponential improvement on standard dithering. Sparse vectors can be obtained by random sparsification techniques that randomly mask the input vectors and preserve a constant number of coordinates (Wangni et al., 2018; Konečný & Richtárik, 2018; Stich et al., 2018; Mishchenko et al., 2019; Vogels et al., 2019). There is also a line of work (Horváth et al., 2019; Basu et al., 2019) which propose to combine sparsification and quantization to obtain a more aggressive combined effect.

**Client Sampling/Selection Methods.** In the situation where partial participation is desired and a budget on the number of participating clients is applied, *a careful selection of the participating clients can lead to better communication complexity, and hence faster training.* In other words, some clients will have "more informative" updates than others in any given communication round, and thus the training procedure will benefit from capitalizing on this fact by ignoring some of the worthless updates (see Figure 1). We refer the readers to Section 4.1 for discussions on existing client sampling methods in FL and their limitations.

## 1.2 Contributions

We address the communication bandwidth issues appearing in FL by designing a *principled optimal client sampling scheme with client privacy and system practicality in mind.* We show that the ideas presented in the previous works on efficient sampling (Horváth & Richtárik, 2019) and sparsification (Wang et al., 2018; Wangni et al., 2018) can be adapted to be compatible with FL and can be used to construct a principled *optimal client sampling scheme* which is capable of identifying the most informative clients in any given communication round. Our contributions can be summarized as follows:

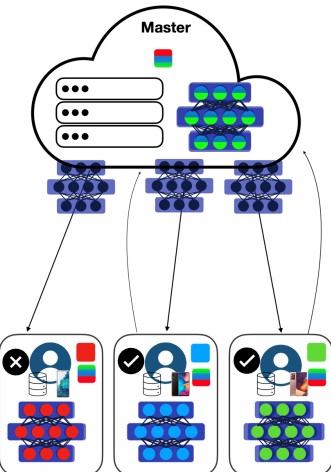

Figure 1: Optimal client sampling: in each communication round, all participating clients compute their updates, but only the ones with "important" updates communicate back to the master.

- Inspired by Horváth & Richtárik (2019), we propose an *adaptive partial participation strategy for reducing communication in FL*. This strategy relies on a careful selection of clients that are allowed to communicate their updates back to the master in any given communication round, which then translates to a reduction in the number of communicated bits. We obtain this strategy by properly applying the sampling procedure from Horváth & Richtárik (2019) to the FL framework.

- Specifically, building upon the importance sampling results in Horváth & Richtárik (2019, Lemma 1), we obtain an *optimal* adaptive client sampling procedure in the sense that it minimizes the variance of the master update for any budget $m$ on the number of participating clients, which generalizes the theoretical results in Zhao & Zhang (2015) that only applies to $m = 1$.

- Inspired by the greedy algorithm from Wangni et al. (2018, Algorithm 3) which was originally designed for gradient sparsification, we obtain an approximation to our optimal sampling strategy which only requires aggregation, fulfilling two core privacy requirements of FL: to our knowledge, our method is the first principled importance client sampling strategy that is compatible with both *secure aggregation* and *stateless clients*.

- Our optimal sampling method is orthogonal to and hence compatible with existing approaches to communication reduction such as communication compression and/or local updates (cf. Section 3.2).

- We provide convergence guarantees for our approach with Distributed `SGD` (`DSGD`) and Federated Averaging (`FedAvg`), relaxing a number of strong assumptions employed in prior works. We show both theoretically and empirically that the performance of our approach is superior to uniform sampling and can be close to full participation.

- We show both theoretically and empirically that our approach allows for *larger learning rates* than the baseline which performs uniform client sampling, which results in *better communication complexity* and hence *faster convergence*.

## 1.3  Organization of the Paper

Section 2 describes the proposed optimal client sampling strategy for reducing the communication bottleneck in federated learning. Section 3 provides convergence analyses for `DSGD` and `FedAvg` with our optimal client sampling scheme in both convex and non-convex settings. Section 4 reviews prior works that are closely or broadly related to our proposed method. Section 5 empirically evaluates our optimal client sampling method on standard federated datasets. Section 6 summarizes the paper and lists some directions for future work.

## 2 Smart Client Sampling for Reducing Communication

This section describes the proposed optimal client sampling strategy for reducing the communication bottleneck in federated learning.

Before proceeding with our theory, we provide an intuition by discussing the problem setting and introducing the *arbitrary sampling* paradigm. In FL, each client $i$ participating in round $k$ computes an update vector $\mathbf{U}_i^k \in \mathbb{R}^d$. For simplicity and ease of exposition, we assume that all clients $i \in [n] := \{1, 2, \ldots, n\}$ are available in each round[1]. In our framework, only a subset of clients communicates their updates to the master node in each communication round in order to reduce the number of transmitted bits.

In order to provide an analysis in this framework, we consider a general partial participation framework (Horváth & Richtárik, 2020), where we assume that the subset of participating clients is determined by an arbitrary random set-valued mapping $\mathbb{S}$ (i.e., a "sampling") with values in $2^{[n]}$. A sampling $\mathbb{S}$ is uniquely defined by assigning probabilities to all $2^n$ subsets of $[n]$. With each sampling $\mathbb{S}$ we associate a *probability matrix* $\mathbf{P} \in \mathbb{R}^{n \times n}$ defined by $\mathbf{P}_{ij} := \mathrm{Prob}(\{i, j\} \subseteq \mathbb{S})$. The *probability vector* associated with $\mathbb{S}$ is the vector composed of the diagonal entries of $\mathbf{P}$: $p = (p_1, \ldots, p_n) \in \mathbb{R}^n$, where $p_i := \mathrm{Prob}(i \in \mathbb{S})$. We say that $\mathbb{S}$ is *proper* if $p_i > 0$ for all $i$. It is easy to show that $b := \mathrm{E}\left[|\mathbb{S}|\right] = \mathrm{Trace}\left(\mathbf{P}\right) = \sum_{i=1}^n p_i$, and hence $b$ can be seen as the expected number of clients participating in each communication round. Given parameters $p_1, \ldots, p_n \in [0, 1]$, consider a random set $\mathbb{S} \subseteq [n]$ generated as follows: for each $i \in [n]$, we include $i$ in $\mathbb{S}$ with probability $p_i$. This is called *independent sampling*, since the event $i \in \mathbb{S}$ is independent of $j \in \mathbb{S}$ for any $i \neq j$.

While our client sampling strategy can be adapted to essentially any underlying learning method, we give details here for DSGD as an illustrative example, where the master update in each communication round is of the form

$$x^{k+1} = x^k - \eta^k \mathbf{G}^k \quad \text{with} \quad \mathbf{G}^k := \sum_{i \in S^k} \frac{w_i}{p_i^k} \mathbf{U}_i^k, \tag{2}$$

where $S^k \sim \mathbb{S}^k$ and $\mathbf{U}_i^k = g_i^k$ is an unbiased estimator of $\nabla f_i(x^k)$. The scaling factor $\frac{1}{p_i^k}$ is necessary in order to obtain an unbiased estimator of the true update, i.e., $\mathrm{E}_{S^k}\left[\mathbf{G}^k\right] = \sum_{i=1}^n w_i \mathbf{U}_i^k$.

### 2.1 Optimal Client Sampling

A simple observation is that the variance of our gradient estimator $\mathbf{G}^k$ can be decomposed into

$$\mathrm{E}\left[\left\|\mathbf{G}^k - \nabla f(x^k)\right\|^2\right] = \mathrm{E}\left[\left\|\mathbf{G}^k - \sum_{i=1}^n w_i \mathbf{U}_i^k\right\|^2\right] + \mathrm{E}\left[\left\|\sum_{i=1}^n w_i \mathbf{U}_i^k - \nabla f(x^k)\right\|^2\right], \tag{3}$$

where the second term on the right-hand side is independent of the sampling procedure, and the first term is zero if every client sends its update (i.e., if $p_i^k = 1$ for all $i$). In order to provide meaningful results, we restrict the expected number of clients to communicate in each round by bounding $b^k := \sum_{i=1}^n p_i^k$ by some positive integer $m \leq n$. This raises the following question: *What is the sampling procedure that minimizes* (3) *for any given $m$?*

To answer this question, we connect Equation (3) to previous works on importance sampling (Horváth & Richtárik, 2019) and gradient sparsification (Wangni et al., 2018; Wang et al., 2018)[2]. Despite difference in motivation, these works solve up to a scale the equivalent mathematical problem, based on which we answer the aforementioned question by the following technical lemma (see Appendix A for a proof):

**Lemma 1.** *(Generalization of Horváth & Richtárik (2019, Lemma 1)) Let $\zeta_1, \zeta_2, \ldots, \zeta_n$ be vectors in $\mathbb{R}^d$ and $w_1, w_2, \ldots, w_n$ be non-negative real numbers such that $\sum_{i=1}^n w_i = 1$. Define $\zeta := \sum_{i=1}^n w_i \zeta_i$. Let $S$ be a proper sampling. If $v \in \mathbb{R}^n$ is such that*

$$\mathbf{P} - pp^\top \preceq \mathbf{Diag}(p_1 v_1, p_2 v_2, \ldots, p_n v_n), \tag{4}$$

---

[1]This is not a limiting factor, as all presented theory can be easily extended to the case of partial participation with an arbitrary proper sampling distribution. See Appendix E for a proof sketch.

[2]Wangni et al. (2018) consider a slightly different problem, where they minimize the communication budget with constraints on the variance.

---

**Algorithm 1** Optimal Client Sampling (`OCS`).

1: **Input:** expected batch size $m$
2: each client $i$ computes a local update $\mathbf{U}_i^k$ (in parallel)
3: each client $i$ sends the norm of its update $u_i^k = w_i \left\| \mathbf{U}_i^k \right\|$ to the master (in parallel)
4: master computes optimal probabilities $p_i^k$ using equation (7)
5: master broadcasts $p_i^k$ to all clients
6: each client $i$ sends its update $\frac{w_i}{p_i^k} \mathbf{U}_i^k$ to the master with probability $p_i^k$ (in parallel)

---

*then*

$$\mathrm{E} \left[ \left\| \sum_{i \in S} \frac{w_i \zeta_i}{p_i} - \tilde{\zeta} \right\|^2 \right] \leq \sum_{i=1}^{n} w_i^2 \frac{v_i}{p_i} \left\| \zeta_i \right\|^2, \tag{5}$$

*where the expectation is taken over $S$. Whenever (4) holds, it must be the case that $v_i \geq 1 - p_i$.*

It turns out that given probabilities $\{p_i\}$, among all samplings $S$ satisfying $p_i = \mathrm{Prob}(i \in S)$, the independent sampling (i.e., $p_{ij} = \mathrm{Prob}(i, j \in S) = \mathrm{Prob}(i \in S) \, \mathrm{Prob}(j \in S) = p_i p_j$) minimizes the left-hand side of (5). This is due to two nice properties: a) any independent sampling admits the optimal choice of $v$, i.e., $v_i = 1 - p_i$ for all $i$, and b) (5) holds as equality for independent sampling. In the context of our method, these properties can be written as

$$\mathrm{E} \left[ \left\| \mathbf{G}^k - \sum_{i=1}^{n} w_i \mathbf{U}_i^k \right\|^2 \right] = \mathrm{E} \left[ \sum_{i=1}^{n} w_i^2 \frac{1 - p_i^k}{p_i^k} \left\| \mathbf{U}_i^k \right\|^2 \right]. \tag{6}$$

It now only remains to find the parameters $\{p_i^k\}$ defining the optimal independent sampling, i.e., one that minimizes (6) subject to the constraints $0 \leq p_i^k \leq 1$ and $b^k := \sum_{i=1}^{n} p_i^k \leq m$. It turns out that this problem has the following closed-form solution (see Appendix B for a proof):

$$p_i^k = \begin{cases} (m + l - n) \dfrac{\left\| \tilde{U}_i^k \right\|}{\sum_{j=1}^{l} \left\| \tilde{U}_{(j)}^k \right\|}, & \text{if } i \notin A^k \\ 1, & \text{if } i \in A^k \end{cases}, \tag{7}$$

where $\tilde{U}_i^k := w_i \mathbf{U}_i^k$, and $\left\| \tilde{U}_{(j)}^k \right\|$ is the $j$-th smallest value in $\{ \| \tilde{U}_i^k \| \}_{i=1}^{n}$, $l$ is the largest integer for which $0 < m + l - n \leq \sum_{i=1}^{l} \| \tilde{U}_{(i)}^k \| / \| \tilde{U}_{(l)}^k \|$ (note that this inequality always holds for $l = n - m + 1$), and $A^k$ contains indices $i$ such that $\| \tilde{U}_i^k \| \geq \left\| \tilde{U}_{(l+1)}^k \right\|$. We summarize this procedure in Algorithm 1. Intuitively, our method can be thought of as uniform sampling with $\tilde{m} \in [m, n]$ effective sampled clients, while only $m$ clients are actually sampled in expectation, which indicates that it cannot be worse than uniform sampling and can be as good as full participation. The actual value of $\tilde{m}$ depends on the updates.

**Remark 2** (Optimality). *Optimizing the left-hand side of (5) does not guarantee the proposed sampling to be optimal with respect to the right-hand side of (5) in the general case. For this to hold, our sampling needs to be independent, which is not a very restrictive condition, especially considering that enforcing independent sampling across clients accommodates the privacy requirements of FL. In addition, since (5) is tight, our sampling is optimal if one is allowed to communicate only norms (i.e., one float per client) as extra information. We stress that requiring optimality with respect to the left-hand side of (5) in the full general case is not practical, as it cannot be obtained without revealing, i.e., communicating, all clients' full updates to the master.*

## 2.2 Ensuring Compatibility with Secure Aggregation and Stateless Clients

In the case $l = n$, the optimal probabilities $p_i^k = m \| \tilde{U}_i^k \| / \sum_{j=1}^{n} \| \tilde{U}_j^k \|$ can be computed easily: the master aggregates the norm of each update and then sends the sum back to the clients. However, if $l < n$, in order

---

**Algorithm 2** Approximate Optimal Client Sampling (`AOCS`).

---
1: **Input:** expected batch size $m$, maximum number of iteration $j_{\max}$
2: each client $i$ computes an update $\mathbf{U}_i^k$ (in parallel)
3: each client $i$ sends the norm of its update $u_i^k = w_i \left\|\mathbf{U}_i^k\right\|$ to the master (in parallel)
4: master aggregates $u^k = \sum_{i=1}^n u_i^k$
5: master broadcasts $u^k$ to all clients
6: each client $i$ computes $p_i^k = \min\{\frac{mu_i^k}{u^k}, 1\}$ (in parallel)
7: **for** $j = 1, \cdots, j_{max}$ **do**
8:     each client $i$ sends $t_i^k = (1, p_i^k)$ to the master if $p_i^k < 1$; else sends $t_i^k = (0, 0)$ (in parallel)
9:     master aggregates $(I^k, P^k) = \sum_{i=1}^n t_i^k$
10:     master computes $C^k = \frac{m-n+I^k}{P^k}$
11:     master broadcasts $C^k$ to all clients
12:     each client $i$ recalibrates $p_i^k = \min\{C^k p_i^k, 1\}$ if $p_i^k < 1$ (in parallel)
13:     **if** $C^k \le 1$ **then**
14:       break
15:     **end if**
16: **end for**
17: each clients $i$ sends its update $\frac{w_i}{p_i^k}\mathbf{U}_i^k$ to master with probability $p_i^k$ (in parallel)

---

to compute optimal probabilities, the master would need to identify the norm of every update and perform partial sorting, which can be computationally expensive and also violates the client privacy requirements in FL, i.e., one cannot use the secure aggregation protocol where the master only sees the sum of the updates. Therefore, we create an algorithm for approximately solving this problem, which only requires to perform aggregation at the master node without compromising the privacy of any client. The construction of this algorithm is built upon the greedy algorithm from Wangni et al. (2018, Algorithm 3) which was originally designed for gradient sparsification but solves up to a scale an equivalent mathematical problem. We first set $\tilde{p}_i^k = m\|\tilde{U}_i^k\|/\sum_{j=1}^n \|\tilde{U}_j^k\|$ and $p_i^k = \min\{\tilde{p}_i^k, 1\}$. In the ideal situation where every $\tilde{p}_i^k$ equals the optimal solution (7), this would be sufficient. However, due to the truncation operation, the expected number of sampled clients $b^k = \sum_{i=1}^n p_i^k \le \sum_{i=1}^n m\|\tilde{U}_i^k\|/\sum_{j=1}^n \|\tilde{U}_j^k\| = m$ can be strictly less than $m$ if $\tilde{p}_i^k > 1$ holds true for at least one $i$. Hence, we employ an iterative procedure to fix this gap by rescaling the probabilities which are smaller than 1, as summarized in Algorithm 2. This algorithm is much easier to implement and computationally more efficient on parallel computing architectures. In addition, it only requires a secure aggregation procedure on the master, which is essential in privacy preserving FL, and thus it is compatible with existing FL software and hardware.

**Remark 3** (Extra communications in Algorithm 2). *We acknowledge that Algorithm 2 brings extra communication costs, as it requires all clients to send the norms of their updates $u_i^k$'s and probabilities $p_i^k$'s in each round. However, since these are single floats, this only costs $\mathcal{O}(j_{\max})$ extra floats for each client. Picking $j_{\max} = \mathcal{O}(1)$, this is negligible for large models of size $d$. We also acknowledge that engaging in multiple synchronous rounds of communication (as in Algorithm 2) can be a bottleneck (Huba et al., 2022). This is not an issue in our work, as we focus on reducing the total communication cost. However, Algorithm 2 may be less useful under other setups or metrics.*

**Remark 4** (Fairness). *Based on our sampling strategy, it might be tempting to assume that the obtained solution could exhibit fairness issues. In our convergence analyses below, we show that this is not the case, as our proposed methods converge to the optimal solution of the original problem. Hence, as long as the original objective has no inherent issue with fairness, our method does not exhibit any fairness issues. Besides, our algorithm can be used in conjunction with other "more fair" objectives, e.g., Tilted ERM (Li et al., 2021), if needed.*

## 3 Convergence Guarantees

This section provides convergence analyses for `DSGD` and `FedAvg` with our optimal client sampling scheme in both convex and non-convex settings. We compare the convergence results of our scheme with those of full participation and independent uniform sampling with sample size $m$. We match the forms of our convergence bounds to those of the existing bounds in the literature to make them directly comparable. We do not compare the sample complexities of these methods, as such comparisons would be difficult due to their dependence on the actual updates which are unknown in advance and do not follow a specific distribution in general.

We use standard assumptions (Karimi et al., 2016), assuming throughout that $f$ has a unique minimizer $x^\star$ with $f^\star = f(x^\star) > -\infty$ and $f_i$'s are $L$-smooth, i.e., $f_i$'s have $L$-Lipschitz continuous gradients. We first define convex functions and $L$-smooth functions.

**Definition 5** (Convexity). $f : \mathbb{R}^d \to \mathbb{R}$ *is $\mu$-strongly convex with $\mu > 0$ if*

$$f(y) \geq f(x) + \langle \nabla f(x), y - x \rangle + \frac{\mu}{2} \|y - x\|^2, \quad \forall x, y \in \mathbb{R}^d. \tag{8}$$

$f : \mathbb{R}^d \to \mathbb{R}$ *is convex if it satisfies* (8) *with $\mu = 0$.*

**Definition 6** (Smoothness). $f : \mathbb{R}^d \to \mathbb{R}$ *is $L$-smooth if*

$$\|\nabla f(x) - \nabla f(y)\| \leq L \|x - y\|, \quad \forall x, y \in \mathbb{R}^d. \tag{9}$$

We now state standard assumptions of the gradient oracles for `DSGD` and `FedAvg`.

**Assumption 7** (Gradient oracle for `DSGD`). *The stochastic gradient estimator $g_i^k = \nabla f_i(x^k) + \xi_i^k$ of the local gradient $\nabla f_i(x^k)$, for each round $k$ and all $i = 1, \ldots, n$, satisfies*

$$\mathrm{E}\left[\xi_i^k\right] = 0 \tag{10}$$

*and*

$$\mathrm{E}\left[\left\|\xi_i^k\right\|^2 | x_i^k\right] \leq M \left\|\nabla f_i(x^k)\right\|^2 + \sigma^2, \quad \text{for some } M \geq 0. \tag{11}$$

*This further implies that $\mathrm{E}\left[\frac{1}{n}\sum_{i=1}^n g_i^k \mid x^k\right] = \nabla f(x^k)$.*

**Assumption 8** (Gradient oracle for `FedAvg`). *The stochastic gradient estimator $g_i(y_{i,r}^k) = \nabla f_i(y_{i,r}^k) + \xi_{i,r}^k$ of the local gradient $\nabla f_i(y_{i,r}^k)$, for each round $k$, each local step $r = 0, \ldots, R$ and all $i = 1, \ldots, n$, satisfies*

$$\mathrm{E}\left[\xi_{i,r}^k\right] = 0 \tag{12}$$

*and*

$$\mathrm{E}\left[\left\|\xi_{i,r}^k\right\|^2 | y_{i,r}^k\right] \leq M \left\|\nabla f_i(y_{i,r}^k)\right\|^2 + \sigma^2, \quad \text{for some } M \geq 0, \tag{13}$$

*where $y_{i,0}^k = x^k$ and $y_{i,r}^k = y_{i,r-1}^k - \eta_l g_i(y_{i,r}^k)$, for $r = 1, \cdots, R$.*

For non-convex objectives, one can construct counter-examples that would diverge for both `DSGD` and `FedAvg` if the sampling variance is not bounded. Therefore, we need to employ the following standard assumption of local gradients for bounding the sampling variance[3].

**Assumption 9** (Similarity among local gradients). *The gradients of local loss functions $f_i$ satisfy*

$$\sum_{i=1}^n w_i \|\nabla f_i(x) - \nabla f(x)\|^2 \leq \rho, \quad \text{for some } \rho \geq 0. \tag{14}$$

---

[3]This assumption is not required for convex objectives, as one can show that the sampling variance is bounded using smoothness and convexity.

**Remark 10** (Interpretation of Assumption 9). *Some works employ a more restrictive assumption which requires $\|\nabla f_i(x) - \nabla f(x)\| \leq \rho$, $\forall i$, from which Assumption 9 can be derived, since $\sum_{i=1}^{n} w_i = 1$. Therefore, Assumption 9 can be seen as an assumption on similarity among local gradients. Furthermore, this assumption does not require $w_i$'s to be lower-bounded, as clients with $w_i = 0$ will never be sampled and thus can be removed from the objective.*

We now define some important quantities for our convergence analyses.

**Definition 11** (The improvement factor). *We define the improvement factor of optimal client sampling over uniform sampling:*

$$\alpha^k := \frac{\mathrm{E}\left[\left\|\sum_{i \in S^k} \frac{w_i}{p_i^k} \mathbf{U}_i^k - \sum_{i=1}^{n} w_i \mathbf{U}_i^k\right\|^2\right]}{\mathrm{E}\left[\left\|\sum_{i \in U^k} \frac{w_i}{p_i^U} \mathbf{U}_i^k - \sum_{i=1}^{n} w_i \mathbf{U}_i^k\right\|^2\right]}, \tag{15}$$

*where $S^k \sim \mathbb{S}^k$ with $p_i^k$ defined in (7) and $U^k \sim \mathbb{U}$ is an independent uniform sampling with $p_i^U = {}^m/n$. By construction, $0 \leq \alpha^k \leq 1$, as $\mathbb{S}^k$ minimizes the variance term (see Appendix B for a proof). Note that $\alpha^k$ can reach zero in the case where there are at most m non-zero updates. If $\alpha^k = 0$, our method performs as if all updates were communicated. In the worst-case $\alpha^k = 1$, our method performs as if we picked m updates uniformly at random, and one could not do better in theory due to the structure of the updates $\mathbf{U}_i^k$. The actual value of $\alpha^k$ will depend on the updates $\mathbf{U}_i^k$. We also define the relative improvement factor:*

$$\gamma^k := \frac{m}{\alpha^k(n-m) + m} \in \left[\frac{m}{n}, 1\right], \quad k = 0, \ldots, K-1. \tag{16}$$

**Definition 12** (Simplified notation). *For simplicity of notation, we define the following quantities which will be useful for our convergence analyses:*

$$W := \max_{i \in [n]}\{w_i\}, \quad Z_i := f_i(x^\star) - f_i^\star, \quad r^k := x^k - x^\star, \tag{17}$$

*where $f_i^\star$ is the functional value of $f_i$ at its optimum, $Z_i$ represents the mismatch between the local and global minimizer, and $r^k$ captures the distance between the current point and the minimizer of $f$.*

We are now ready to proceed with our convergence analyses. In the following subsections, we provide convergence analyses of specific methods for solving the optimization problem (1). The proofs of the theorems are deferred to Appendices C and D.

### 3.1 Distributed `SGD` (`DSGD`) with Optimal Client Sampling

This subsection presents convergence analyses for `DSGD` (2) with optimal client sampling in both convex and non-convex settings.

**Theorem 13** (`DSGD`, strongly-convex). *Let $f_i$ be $L$-smooth and convex for $i = 1, \ldots, n$. Let $f$ be $\mu$-strongly convex. Suppose that Assumption 7 holds. Choose $\eta^k \in \left(0, \frac{\gamma^k}{(1+WM)L}\right]$. Define*

$$\beta_1 := \sum_{i=1}^{n} w_i^2(2L(1+M)Z_i + \sigma^2), \tag{18}$$

$$\beta_2 := 2L \sum_{i=1}^{n} w_i^2 Z_i. \tag{19}$$

*The iterates of `DSGD` with optimal client sampling (7) satisfy*

$$\mathrm{E}\left[\|r^{k+1}\|^2\right] \leq (1 - \mu\eta^k)\mathrm{E}\left[\|r^k\|^2\right] + (\eta^k)^2\left(\frac{\beta_1}{\gamma^k} - \beta_2\right). \tag{20}$$

**Remark 14** (Interpretation of Theorem 13). *We first look at the best and worst case scenarios. In the best case scenario, we have $\gamma^k = 1$ for all $k$'s. This implies that there is no loss of speed comparing to the method with full participation. It is indeed confirmed by our theory as our obtained recursion recovers the best-known rate of* DSGD *in the full participation regime (Gower et al., 2019, Theorem 3.1). To provide a better intuition, we include a full derivation in this case. To match their (stronger) assumptions, we let $M = 0$ and $w_i = 1/n$. In full participation, we have $\gamma^k = 1$ for all $k$'s. Then, taking the same step size $\eta$ for all $k$ leads to*

$$\mathrm{E}\left[\left\|r^{k+1}\right\|^2\right] \le (1 - \mu\eta)\mathrm{E}\left[\left\|r^k\right\|^2\right] + \eta^2\frac{\sigma^2}{n}. \tag{21}$$

*Applying the above inequality recursively yields*

$$\mathrm{E}\left[\left\|r^K\right\|^2\right] \le (1 - \mu\eta)^K\mathrm{E}\left[\left\|r^0\right\|^2\right] + \eta\frac{\sigma^2}{\mu n}, \tag{22}$$

*which is equivalent to the result in Gower et al. (2019, Theorem 3.1). Similarly, in the worst case, we have $\gamma^k = m/n$ for all $k$'s, which corresponds to uniform sampling with sample size $m$, and our recursion recovers the best-known rate for* DSGD *in this regime. This is expected as (15) implies that every update $\mathbf{U}_i^k$ is equivalent, and thus it is theoretically impossible to obtain a better rate than that of uniform sampling in the worst case scenario. In the general scenario, our obtained recursion sits somewhere between full and uniform partial participation, where the actual position is determined by $\gamma^k$'s which capture the distribution of updates (here gradients) on the clients. For instance, with a larger number of $\gamma^k$'s tending to 1, we are closer to the full participation regime. Similarly, with more $\gamma^k$'s tending to $m/n$, we are closer to the rate of uniform partial participation.*

**Theorem 15** (DSGD, non-convex). *Let $f_i$ be $L$-smooth for $i = 1, \ldots, n$. Suppose that Assumptions 7 and 9 hold. Let $\eta^k$ be the step size and define*

$$\beta^k := \frac{L}{2\gamma^k}\left((1 + M - \gamma^k)W\rho + \sum_{i=1}^n w_i^2\sigma^2\right). \tag{23}$$

*The iterates of* DSGD *with optimal client sampling (7) satisfy*

$$\mathrm{E}\left[f(x^{k+1})\right] \le \mathrm{E}\left[f(x^k)\right] - \eta^k\left(1 - \frac{(1 + M)L}{2\gamma^k}\eta^k\right)\mathrm{E}\left[\left\|\nabla f(x^k)\right\|^2\right] + (\eta^k)^2\beta^k. \tag{24}$$

**Remark 16** (Interpretation of Theorem 15). *The iterate (24) recovers the standard form of the convergence result of* DSGD *for one recursion step in the non-convex setting. Similar to the previous results, this convergence bound sits between the best-known rate of full participation and uniform sampling (Bottou et al., 2018, Theorem 4.8).*

## 3.2 Federated Averaging (FedAvg) with Optimal Client Sampling

Pseudo-code that adapts the standard FedAvg algorithm to our framework is provided in Algorithm 3. This subsection presents convergence analyses for FedAvg with optimal client sampling in both convex and non-convex settings.

**Theorem 17** (FedAvg, strongly-convex). *Let $f_i$ be $L$-smooth and $\mu$-strongly convex for $i = 1, \ldots, n$. Suppose that Assumption 8 holds. Let $\eta^k := R\eta_l^k\eta_g^k$ be the effective step-size and $\eta_g^k \ge \sqrt{\frac{\gamma^k}{\sum_i w_i^2}}$. Choose $\eta^k \in \left(0, \frac{1}{8}\min\left\{\frac{1}{L(2+M/R)}, \frac{\gamma^k}{(1+W(1+M/R))L}\right\}\right]$,*

$$\beta_1^k := \frac{2\sigma^2}{\gamma^k R}\sum_{i=1}^n w_i^2 + 4L\left(\frac{M}{R} + 1 - \gamma^k\right)\sum_{i=1}^n w_i^2 Z_i, \tag{25}$$

$$\beta_2 := 72L^2\left(1 + \frac{M}{R}\right)\sum_{i=1}^n w_i Z_i. \tag{26}$$

---

**Algorithm 3** `FedAvg` with Optimal Client Sampling.

---

1: **Input:** initial global model $x^1$, global and local step-sizes $\eta_g^k$, $\eta_l^k$
2: **for** each round $k = 1, \ldots, K$ **do**
3:      master broadcasts $x^k$ to all clients $i \in [n]$
4:      **for** each client $i \in [n]$ (in parallel) **do**
5:          initialize local model $y_{i,0}^k \leftarrow x^k$
6:          **for** $r = 1, \ldots, R$ **do**
7:              compute mini-batch gradient $g_i(y_{i,r-1}^k)$
8:              update $y_{i,r}^k \leftarrow y_{i,r-1}^k - \eta_l^k g_i(y_{i,r-1}^k)$
9:          **end for**
10:          compute $\mathbf{U}_i^k := \Delta y_i^k = x^k - y_{i,R}^k$
11:          compute $p_i^k$ using Algorithm 1 or 2
12:          send $\frac{w_i}{p_i^k} \Delta y_i^k$ to master with probability $p_i^k$
13:      **end for**
14:      master computes $\Delta x^k = \sum_{i \in S^k} \frac{w_i}{p_i^k} \Delta y_i^k$
15:      master updates global model $x^{k+1} \leftarrow x^k - \eta_g^k \Delta x^k$
16: **end for**

---

*The iterates of* `FedAvg` *($R \geq 2$) with optimal client sampling* (7) *satisfy*

$$\frac{3}{8}\mathrm{E}\left[(f(x^k) - f^\star)\right] \leq \frac{1}{\eta^k}\left(1 - \frac{\mu\eta^k}{2}\right)\mathrm{E}\left[\|r^k\|^2\right] - \frac{1}{\eta^k}\mathrm{E}\left[\|r^{k+1}\|^2\right] + \eta^k\beta_1^k + (\eta^k)^2\beta_2. \tag{27}$$

**Theorem 18** (`FedAvg`, non-convex)**.** *Let $f_i$ be $L$-smooth for all $i = 1, \ldots, n$. Suppose that Assumptions 8 and 9 hold. Let $\eta^k := R\eta_l^k\eta_g^k$ be the effective step-size and $\eta_g^k \geq \sqrt{\frac{5\gamma^k}{4\sum_i w_i^2}}$. Choose $\eta^k \in \left(0, \frac{1}{8L(2+M/R)}\right]$. Define*

$$\beta^k := \left(\frac{\rho}{4} + \frac{\sigma^2}{\gamma^k R}\sum_{i=1}^n w_i^2\right) L. \tag{28}$$

*The iterates of* `FedAvg` *($R \geq 2$) with optimal client sampling* (7) *satisfy*

$$\mathrm{E}\left[f(x^{k+1})\right] \leq \mathrm{E}\left[f(x^k)\right] - \frac{3\eta^k}{8}\left(1 - \frac{10\eta^k L}{3}\right)\mathrm{E}\left[\|\nabla f(x^k)\|^2\right] + \eta^k\frac{\rho}{8} + (\eta^k)^2\beta^k. \tag{29}$$

**Remark 19** (Interpretation of Theorems 17 and 18)**.** *The convergence guarantees from Theorems 17 and 18 sit somewhere between those for full and uniform partial participation. The actual position is again determined by the distribution of the updates which are linked to $\gamma^k$'s. In the edge cases, i.e., $\gamma^k = 1$ (best case) or $\gamma^k = m/n$ (worst case), we recover the state-of-the-art complexity guarantees provided in (Karimireddy et al., 2019, Theorem I) in both regimes. Note that our results are slightly more general, as Karimireddy et al. (2019) assumes $M = 0$ and $w_i = 1/n$.*

## 4 Related Work

This section reviews prior works that are closely or broadly related to our proposed method.

### 4.1 Importance Client Sampling in Federated Learning

Several recent works have studied efficient importance client sampling methods in FL (Cho et al., 2020; Nguyen et al., 2020; Ribero & Vikalo, 2020; Lai et al., 2021; Luo et al., 2022). Unfortunately, none of

these methods is principled, as they rely on heuristics, historical losses, or partial information, which can be seen as proxies for our optimal client sampling. Furthermore, they violate at least one of the core privacy requirements of FL (secure aggregation and/or stateless clients). Specifically, the client selection strategy proposed by Lai et al. (2021) is based on the heuristic of system and statistical utility of clients, which reveals the identity of clients; Ribero & Vikalo (2020) propose to model the progression of the model parameters by an Ornstein-Uhlenbeck process based on partial information, where the master needs to process the raw update from each client. The work of Cho et al. (2020) biases client selection towards clients with higher local losses, which reveals the state of each individual client.

In contrast, our proposed method is the first *principled optimal client sampling strategy* in the sense that it minimizes the variance of the master update and is *compatible with core privacy requirements of FL*. We note that the client sampling/selection techniques mentioned in this section could be made compatible with our framework presented in Section 2, but they would not lead to the optimal method as they are only proxies for optimal sampling.

### 4.2 Importance Sampling in Stochastic Optimization

Importance sampling methods for optimization have been studied extensively in the last few years in several contexts, including convex optimization and deep learning. `LASVM` developed in Bordes et al. (2005) is an online algorithm that uses importance sampling to train kernelized support vector machines. The first importance sampling for randomized coordinate descent methods was proposed in the seminal paper of Nesterov (2012). It was showed by Richtárik & Takáč (2014) that the proposed sampling is optimal. Later, several extensions and improvements followed, e.g., Shalev-Shwartz & Zhang (2014); Lin et al. (2014); Fercoq & Richtárik (2015); Qu et al. (2015); Allen-Zhu et al. (2016); Stich et al. (2017). Another branch of work studies sample complexity. In Needell et al. (2014); Zhao & Zhang (2015), the authors make a connection with the variance of the gradient estimates of `SGD` and show that the optimal sampling distribution is proportional to the per-sample gradient norm. However, obtaining this distribution is as expensive as computing the full gradient in terms of computation, and thus it is not practical. For simpler problems, one can sample proportionally to the norms of the inputs, which can be linked to the Lipschitz constants of the per-sample loss function for linear and logistic regression. For instance, it was shown by Horváth & Richtárik (2019) that static optimal sampling can be constructed even for mini-batches and the probability is proportional to these Lipschitz constants under the assumption that these constants of the per-sample loss function are known. Unfortunately, importance measures such as smoothness of the gradient are often hard to compute/estimate for more complicated models such as those arising in deep learning, where most of the importance sampling schemes are based on heuristics. For instance, a manually designed sampling scheme was proposed in Bengio et al. (2009). It was inspired by the perceived way that human children learn; in practice, they provide the network with examples of increasing difficulty in an arbitrary manner. In a diametrically opposite approach, it is common for deep embedding learning to sample hard examples because of the plethora of easy non-informative ones (Schroff et al., 2015; Simo-Serra et al., 2015). Other approaches use history of losses for previously seen samples to create the sampling distribution and sample either proportionally to the loss or based on the loss ranking (Schaul et al., 2015; Loshchilov & Hutter, 2015). Katharopoulos & Fleuret (2018) propose to sample based on the gradient norm of a small uniformly sampled subset of samples.

Although our proposed optimal sampling method adapts and extends the importance sampling results from Horváth & Richtárik (2019) to the distributed setting of FL, it does not suffer from any of the limitations discussed above, since the motivation of our work is to *reduce communication* rather than reduce computation. In particular, our method allows for any budge $m < n$ on the number of participating clients, which generalizes the theoretical results from Zhao & Zhang (2015) which only applies to the case $m = 1$.

## 5 Experiments

This section empirically evaluates our optimal client sampling method on standard federated datasets from LEAF (Caldas et al., 2018).

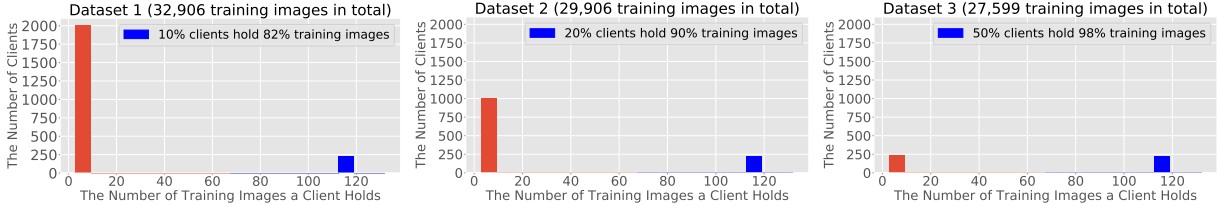

Figure 2: Distributions of the three modified Federated EMNIST training sets.

## 5.1 Setup

We compare our method with 1) full participation where all available clients participate in each round; and 2) the baseline where participating clients are sampled uniformly from available clients in each round. We chose not to compare with other client sampling methods, as such comparisons would be unfair. This is because they violate the privacy requirements of FL: our method is the only importance client sampling strategy that is deployable to real-world FL systems (cf. Section 4.1).

We simulate the cross-device FL distributed setting and train our models using TensorFlow Federated (TFF). We conclude our evaluations using `FedAvg` with Algorithm 2, as it supports stateless clients and secure aggregation[4]. We extend the TFF implementation of `FedAvg` to fit our framework. For all three methods, we report validation accuracy and (local) training loss as a function of the number of communication rounds and the number of bits communicated from clients to the master[5] . Each figure displays the mean performance with standard deviation over 5 independent runs for each of the three compared methods. For a fair comparison, we use the same random seed for all three methods in a single run and vary random seeds across different runs. Detailed experimental settings and extra results can be found in Appendices F.1 and F.2. Our code together with datasets can be found at `https://github.com/SamuelHorvath/FL-optimal-client-sampling`.

## 5.2 Federated EMNIST Dataset

We first evaluate our method on the Federated EMNIST (FEMNIST) image dataset for image classification. Since it is a well-balanced dataset with data of similar quality on each client, we modify its training set by removing some images from some clients, in order to better simulate the conditions in which our proposed method brings significant theoretical improvements. As a result, we produce three unbalanced training sets[6] as summarized in Figure 2. We use the same CNN model as the one used in (McMahan et al., 2017). For validation, we use the unchanged EMNIST validation set, which consists of $40,832$ images. In each communication round, $n = 32$ clients are sampled uniformly from the client pool, each of which then performs several SGD steps on its local training images for 1 epoch with batch size 20. For partial participation, the expected number of clients allowed to communicate their updates back to the master is set to $m \in \{3,6\}$. We use vanilla SGD optimizers with constant step sizes for both clients and the master, with $\eta_g = 1$ and $\eta_l$ tuned on a holdout set. For full participation and optimal sampling, it turns out that $\eta_l = 2^{-3}$ is the optimal local step size for all three datasets. For uniform sampling, the optimal is $\eta_l = 2^{-5}$ for Dataset 1 and $\eta_l = 2^{-4}$ for Datasets 2 and 3. We set $j_{\max} = 4$ and include the extra communication costs in our results. The main results are shown in Figures 3, 4 and 5.

---

[4]We compared the results of Algorithms 1 and 2 for all experiments as a subroutine. Their results are identical, so we only show results for Algorithm 2 and argue that the performance loss caused by its approximation is negligible.

[5]The communication from the master to clients is not considered as a bottleneck and thus not included in the results. This is a standard consideration for distributed systems, as one-to-many communication primitives (i.e., from the master to clients) are several orders of magnitude faster than many-to-one communication primitives (i.e., from clients to the master). This gap is further exacerbated in FL due to the large number of clients and slow client connections.

[6]The aim of creating various unbalanced datasets is to show that optimal sampling has more performance gains over uniform sampling on more unbalanced datasets, since $\alpha^k$'s (defined in Equation (15)) are more likely to be close to zero in this case. These datasets are created using the following procedure. Let $s \in (0,1)$ and $a,b \in \mathbb{N}_+$ with $a < b$. For a given client with $n_c$ examples, we keep this client unchanged if $n_c \leq a$ or $n_c \geq b$, otherwise we remove this client from the dataset with probability $s$ or only keep $a$ randomly sampled examples in this client with probability $1 - s$.

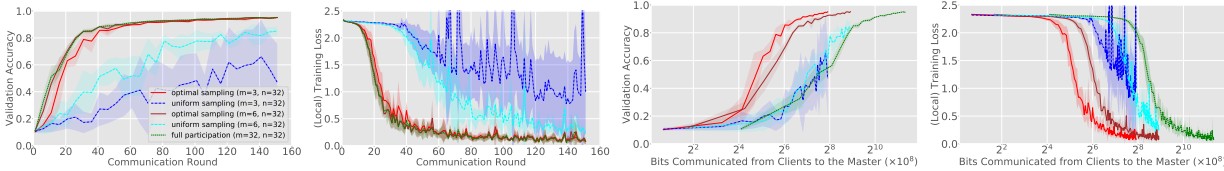

Figure 3: (FEMNIST Dataset 1, $n = 32$) Validation accuracy and (local) training loss as a function of the number of communication rounds and the number of bits communicated from clients to the master.

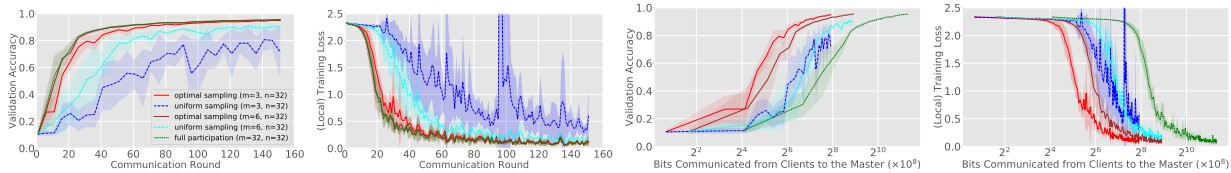

Figure 4: (FEMNIST Dataset 2, $n = 32$) Validation accuracy and (local) training loss as a function of the number of communication rounds and the number of bits communicated from clients to the master.

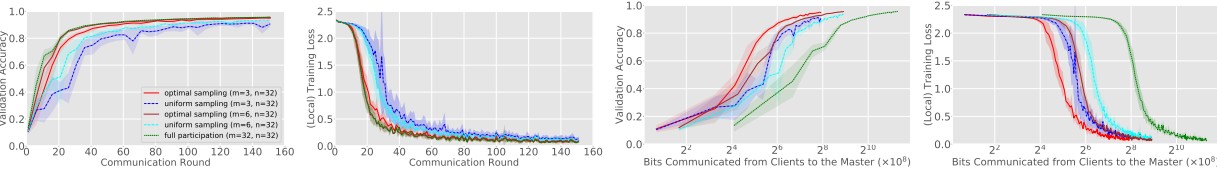

Figure 5: (FEMNIST Dataset 3, $n = 32$) Validation accuracy and (local) training loss as a function of the number of communication rounds and the number of bits communicated from clients to the master.

### 5.3 Shakespeare Dataset

We also evaluate our method on the Shakespeare text dataset for next character prediction. Unlike in the FEMNIST experiments, we do not change the number of examples held by each client in this dataset. The vocabulary set for this task consists of 86 unique characters. The dataset contains 715 clients, each corresponding to a character in Shakespeare's plays. We divide the text into batches such that each batch contains 8 example sequences of length 5. We use a two-hidden-layer GRU model with 256 units in each hidden layer. We set $n \in \{32, 128\}$, $m \in \{2, 4, 6, 12\}$, $j_{max} = 4$, and run several SGD steps for 1 epoch on each client's local dataset in every communication round. We use vanilla SGD optimizers with constant step sizes, with $\eta_g = 1$ and $\eta_l$ tuned on a holdout set. For full participation and optimal sampling, it turns out that the optimal is $\eta_l = 2^{-2}$. For uniform sampling, the optimal is $\eta_l = 2^{-3}$. The main results are shown in Figures 6 and 7.

### 5.4 Discussions

As predicted by our theory, the performance of `FedAvg` with our proposed optimal client sampling strategy is in between that with full and uniform partial participation. For all datasets, the optimal sampling strategy performs slightly worse than but is still competitive with the full participation strategy in terms of the number of communication rounds: it almost reached the performance of full participation while only less than 10% of the available clients communicate their updates back to the master (in the cases $m = 2, 3$). As we increase the expected number $m$ of sampled clients, the performance of optimal sampling increases accordingly, which is consistent with our theory (e.g., Theorem 18) and with the observations from Yang et al. (2021), and quickly becomes almost identical to that of full participation. Note that the uniform sampling strategy performs significantly worse, which indicates that a careful choice of sampling probabilities can go a long way towards closing the gap between the performance of naive uniform sampling and full participation.

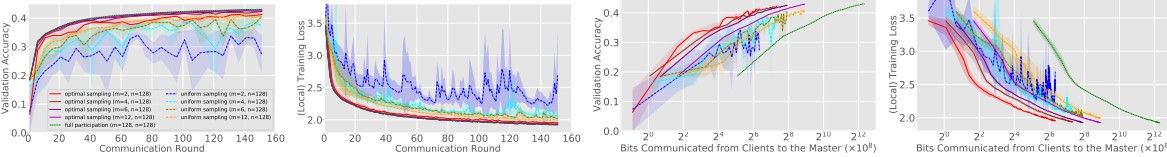

Figure 6: (Shakespeare Dataset, $n = 32$) Validation accuracy and (local) training loss as a function of the number of communication rounds and the number of bits communicated from clients to the master.

Figure 7: (Shakespeare Dataset, $n = 128$) Validation accuracy and (local) training loss as a function of the number of communication rounds and the number of bits communicated from clients to the master.

Also, it can be seen that the performances of our optimal client sampling strategy with $m = 6$ and $m = 12$ match the performances of full participation in the cases $n = 32$ and $n = 128$, respectively, in terms of the number of communication rounds. We therefore conjecture that $m = \mathcal{O}(\sqrt{n})$ is sufficient for our optimal client sampling strategy to obtain identical validation accuracy to that of full participation in terms of the number of communication rounds.

More importantly, and this was the main motivation of our work, our optimal sampling strategy is significantly better than both the uniform sampling and full participation strategies when we compare validation accuracy as a function of the number of bits communicated from clients to the master. For instance, on FEMNIST Dataset 1 (Figure 3), while our optimal sampling approach with $m = 3$ reached around 85% validation accuracy after $2^6 \times 10^8$ communicated bits, neither the full sampling strategy nor the uniform sampling strategy with $m = 3$ is able to exceed 40% validation accuracy within the same communication budget. Indeed, to reach the same 85% validation accuracy, full participation approach needs to communicate more than $2^9 \times 10^8$ bits, i.e., 8× more, and uniform sampling approach needs to communicate about the same number of bits as full participation or even more. The results for FEMNIST Datasets 2 and 3 and for the Shakespeare dataset are of a similar qualitative nature, showing that these conclusions are robust across the datasets considered.

Finally, it is also worth noting that the empirical results from Sections 5.2 and 5.3 confirm that our optimal sampling strategy allows for larger step sizes than uniform sampling, as the hyperparameter search returns larger step sizes $\eta_l$ for optimal sampling than for uniform sampling.

In Appendix G, we present an additional experiment on the Federated CIFAR100 dataset from LEAF. The Federated CIFAR100 dataset is a balanced dataset, where every client holds the same number of training images. In this setting, letting all clients perform 1 epoch of local training means that all clients have the same number of local steps in each round. We show that our optimal client sampling scheme still achieves better performance than uniform sampling on this balanced dataset.

## 6 Conclusion and Future Work

In this work, we have proposed a principled optimal client sampling strategy to address the communication bottleneck issue of federated learning. Our optimal client sampling can be computed using a closed-form formula by aggregating only the norms of the updates. Furthermore, our method is the first principled importance client sampling strategy that is compatible with stateless clients and secure aggregation. We have obtained convergence guarantees for our method with `DSGD` and `FedAvg` with relaxed assumptions, and have performed empirical evaluations of our method on federated datasets from the LEAF database. The

empirical results show that our method is superior to uniform sampling and close to full participation, which corroborates our theoretical analysis. We believe that our proposed optimal client sampling scheme will be useful in reducing communication costs in real-world FL systems.

Some directions for future work are as follows:

- A straightforward extension would be to combine our proposed optimal sampling approach with communication compression methods to further reduce the sizes of communicated updates.

- In the settings where the communication latency is high, our proposed method may not be effective in reducing the real communication time. It would be interesting to extend our optimal client sampling strategy to take into account the constraints of local clients (e.g., computational speed, network bandwidth, and communication latency).

### Acknowledgments

We thank Jakub Konečný for helpful discussions and comments. Most of the work was done when WC was a research intern at KAUST and when SH was a PhD student at KAUST.

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

## A    Proof of Lemma 1

*Proof.* Our proof technique can be seen as an extended version of that in (Horváth & Richtárik, 2019). Let $1_{i\in S} = 1$ if $i \in S$ and $1_{i\in S} = 0$ otherwise. Likewise, let $1_{i,j\in S} = 1$ if $i, j \in S$ and $1_{i,j\in S} = 0$ otherwise. Note that $\mathrm{E}\left[1_{i\in S}\right] = p_i$ and $\mathrm{E}\left[1_{i,j\in S}\right] = p_{ij}$. Next, let us compute the mean of $X := \sum_{i\in S} \frac{w_i\zeta_i}{p_i}$:

$$\mathrm{E}\left[X\right] = \mathrm{E}\left[\sum_{i\in S}\frac{w_i\zeta_i}{p_i}\right] = \mathrm{E}\left[\sum_{i=1}^{n}\frac{w_i\zeta_i}{p_i}1_{i\in S}\right] = \sum_{i=1}^{n}\frac{w_i\zeta_i}{p_i}\mathrm{E}\left[1_{i\in S}\right] = \sum_{i=1}^{n}w_i\zeta_i = \tilde{\zeta}.$$

Let $\boldsymbol{A} = [a_1, \ldots, a_n] \in \mathbb{R}^{d\times n}$, where $a_i = \frac{w_i\zeta_i}{p_i}$, and let $e$ be the vector of all ones in $\mathbb{R}^n$. We now write the variance of $X$ in a form which will be convenient to establish a bound:

$$\begin{aligned}
\mathrm{E}\left[\|X - \mathrm{E}\left[X\right]\|^2\right] &= \mathrm{E}\left[\|X\|^2\right] - \|\mathrm{E}\left[X\right]\|^2 \\
&= \mathrm{E}\left[\left\|\sum_{i\in S}\frac{w_i\zeta_i}{p_i}\right\|^2\right] - \left\|\tilde{\zeta}\right\|^2 \\
&= \mathrm{E}\left[\sum_{i,j}\frac{w_i\zeta_i^\top}{p_i}\frac{w_j\zeta_j}{p_j}1_{i,j\in S}\right] - \left\|\tilde{\zeta}\right\|^2 \\
&= \sum_{i,j}p_{ij}\frac{w_i\zeta_i^\top}{p_i}\frac{w_j\zeta_j}{p_j} - \sum_{i,j}w_iw_j\zeta_i^\top\zeta_j \\
&= \sum_{i,j}(p_{ij} - p_ip_j)a_i^\top a_j \\
&= e^\top((\boldsymbol{P} - pp^\top)\circ\boldsymbol{A}^\top\boldsymbol{A})e.
\end{aligned}$$ (30)

Since, by assumption, we have $\boldsymbol{P} - pp^\top \preceq \mathbf{Diag}(p\circ v)$, we can further bound

$$e^\top((\boldsymbol{P} - pp^\top)\circ\boldsymbol{A}^\top\boldsymbol{A})e \leq e^\top(\mathbf{Diag}(p\circ v)\circ\boldsymbol{A}^\top\boldsymbol{A})e = \sum_{i=1}^{n}p_iv_i\left\|a_i\right\|^2.$$

To obtain (5), it remains to combine this with (30). The inequality $v_i \geq 1 - p_i$ follows by comparing the diagonal elements of the two matrices in (4). Consider now the independent sampling. Clearly,

$$\boldsymbol{P} - pp^\top = \begin{bmatrix} p_1(1-p_1) & 0 & \ldots & 0 \\ 0 & p_2(1-p_2) & \ldots & 0 \\ \vdots & \vdots & \ddots & \vdots \\ 0 & 0 & \ldots & p_n(1-p_n) \end{bmatrix} = \mathbf{Diag}(p_1v_1,\ldots,p_nv_n),$$

which implies $v_i = 1 - p_i$. $\qquad\square$

## B    The Improvement Factor for Optimal Client Sampling

By Lemma 1, the independent sampling (which operates by independently flipping a coin and with probability $p_i$ includes element $i$ into $S$) is optimal. In addition, for independent sampling, (5) holds as equality. Thus, letting $\tilde{U}_i^k = w_i\mathbf{U}_i^k$, we have

$$\tilde{\alpha}_{S^k} := \mathrm{E}\left[\left\|\sum_{i\in S^k}\frac{w_i}{p_i^k}\mathbf{U}_i^k - \sum_{i=1}^{n}w_i\mathbf{U}_i^k\right\|^2\right] = \mathrm{E}\left[\left\|\sum_{i\in S^k}\frac{1}{p_i^k}\tilde{U}_i^k - \sum_{i=1}^{n}\tilde{U}_i^k\right\|^2\right] = \mathrm{E}\left[\sum_{i=1}^{n}\frac{1-p_i^k}{p_i^k}\left\|\tilde{U}_i^k\right\|^2\right].$$ (31)

The optimal probabilities are obtained by minimizing (31) w.r.t. $\{p_i^k\}_{i=1}^{n}$ subject to the constraints $0 \leq p_i^k \leq 1$ and $m \geq b^k = \sum_{i=1}^{n}p_i^k$.

**Lemma 20.** *The optimization problem*

$$\min_{\{p_i^k\}_{i=1}^n} \tilde{\alpha}_{S^k}(\{p_i^k\}_{i=1}^n) \quad s.t. \quad 0 \le p_i^k \le 1, \ \forall i = 1, \cdots, n \quad and \quad m \ge \sum_{i=1}^n p_i^k \tag{32}$$

*has the following closed-form solution:*

$$p_i^k = \begin{cases} (m + l - n)\dfrac{\|\tilde{U}_i^k\|}{\sum_{j=1}^l \|\tilde{U}_{(j)}^k\|}, & \text{if } i \notin A^k \\ 1, & \text{if } i \in A^k \end{cases}, \tag{33}$$

*where $\left\|\tilde{U}_{(j)}^k\right\|$ is the $j$-th largest value among the values $\left\|\tilde{U}_1^k\right\|, \left\|\tilde{U}_2^k\right\|, \ldots, \left\|\tilde{U}_n^k\right\|$, $l$ is the largest integer for which $0 < m + l - n \le \dfrac{\sum_{i=1}^l \left\|\tilde{U}_{(i)}^k\right\|}{\left\|\tilde{U}_{(l)}^k\right\|}$ (note that this inequality at least holds for $l = n - m + 1$), and $A^k$ contains indices $i$ such that $\left\|\tilde{U}_i^k\right\| \ge \left\|\tilde{U}_{(l+1)}^k\right\|$.*

*Proof.* This proof uses an argument similar to that in the proof of Lemma 2 in Horváth & Richtárik (2019). We first show that (33) is the solution to the following optimization problem:

$$\min_{\{p_i^k\}_{i=1}^n} \Omega_{S^k}(\{p_i^k\}_{i=1}^n) \coloneqq \mathrm{E}\left[\sum_{i=1}^n \frac{\|\tilde{U}_i^k\|^2}{p_i^k}\right] \quad s.t. \quad 0 \le p_i^k \le 1, \ \forall i = 1, \cdots, n \quad and \quad m \ge \sum_{i=1}^n p_i^k.$$

The Lagrangian of this optimization problem is given by

$$L(\{p_i^k\}_{i=1}^n, \{\lambda_i\}_{i=1}^n, \{u_i\}_{i=1}^n, y) = \Omega_{S^k}(\{p_i^k\}_{i=1}^n) - \sum_{i=1}^n \lambda_i p_i - \sum_{i=1}^n u_i(1 - p_i) - y\left(m - \sum_{i=1}^n p_i^k\right).$$

Since all constraints are linear and the support of $\{p_i^k\}_{i=1}^n$ is convex, the KKT conditions hold. Therefore, the solution (33) can be deduced from the KKT conditions. Now, notice that $\tilde{\alpha}_{S^k}(\{p_i^k\}_{i=1}^n)$ and $\Omega_{S^k}(\{p_i^k\}_{i=1}^n)$ are equal up to a constant $\mathrm{E}\left[\sum_{i=1}^n \left\|\tilde{U}_i^k\right\|^2\right]$:

$$\tilde{\alpha}_{S^k}(\{p_i^k\}_{i=1}^n) = \Omega_{S^k}(\{p_i^k\}_{i=1}^n) - \mathrm{E}\left[\sum_{i=1}^n \left\|\tilde{U}_i^k\right\|^2\right].$$

This indicates that (33) is also the solution to the original optimization problem (32). $\qquad\square$

Plugging the optimal probabilities obtained in (33) into (31) gives

$$\tilde{\alpha}_{S^k}^\star = \mathrm{E}\left[\sum_{i=1}^n \frac{1}{p_i^k}\left\|\tilde{U}_i^k\right\|^2 - \sum_{i=1}^n \left\|\tilde{U}_i^k\right\|^2\right] = \mathrm{E}\left[\frac{1}{m - (n - l)}\left(\sum_{i=1}^l \left\|\tilde{U}_{(i)}^k\right\|\right)^2 - \sum_{i=1}^l \left\|\tilde{U}_{(i)}^k\right\|^2\right].$$

With $m\left\|\tilde{U}_{(n)}^k\right\| \le \sum_{i=1}^n \left\|\tilde{U}_i^k\right\|$, we have

$$\tilde{\alpha}_{S^k}^\star = \mathrm{E}\left[\frac{1}{m}\left(\sum_{i=1}^n \left\|\tilde{U}_i^k\right\|\right)^2 - \sum_{i=1}^n \left\|\tilde{U}_i^k\right\|^2\right] = \mathrm{E}\left[\frac{1}{m}\left(\sum_{i=1}^n \left\|\tilde{U}_i^k\right\|\right)^2\left(1 - m\frac{\sum_{i=1}^n \left\|\tilde{U}_i^k\right\|^2}{\left(\sum_{i=1}^n \left\|\tilde{U}_i^k\right\|\right)^2}\right)\right]$$

$$\le \frac{n - m}{nm}\mathrm{E}\left[\left(\sum_{i=1}^n \left\|\tilde{U}_i^k\right\|\right)^2\right].$$

For independent uniform sampling $U^k \sim \mathbb{U}$ ($p_i^U = \frac{m}{n}$ for all $i$), we have

$$\tilde{\alpha}_{U^k} := \mathrm{E}\left[\left\|\sum_{i \in U^k} \frac{w_i}{p_i^U} \mathbf{U}_i^k - \sum_{i=1}^{n} w_i \mathbf{U}_i^k \right\|^2\right] = \mathrm{E}\left[\sum_{i=1}^{n} \frac{1 - \frac{m}{n}}{\frac{m}{n}} \left\|\tilde{U}_i^k\right\|^2\right] = \frac{n-m}{m} \mathrm{E}\left[\sum_{i=1}^{n} \left\|\tilde{U}_i^k\right\|^2\right].$$

Putting them together gives the improvement factor:

$$\alpha^k := \frac{\tilde{\alpha}_{S^k}^{\star}}{\tilde{\alpha}_{U^k}} = \frac{\mathrm{E}\left[\left\|\sum_{i \in S^k} \frac{w_i}{p_i^k} \mathbf{U}_i^k - \sum_{i=1}^{n} w_i \mathbf{U}_i^k\right\|^2\right]}{\mathrm{E}\left[\left\|\sum_{i \in U^k} \frac{w_i}{p_i^U} \mathbf{U}_i^k - \sum_{i=1}^{n} w_i \mathbf{U}_i^k\right\|^2\right]} \leq \frac{\mathrm{E}\left[\left(\sum_{i=1}^{n} \left\|\tilde{U}_i^k\right\|\right)^2\right]}{n \mathrm{E}\left[\sum_{i=1}^{n} \left\|\tilde{U}_i^k\right\|^2\right]} \leq 1.$$

The upper bound is attained when all $\left\|\tilde{U}_i^k\right\|$ are identical. Note that the lower bound $0$ can also be attained in the case where the number of non-zero updates is at most $m$. These considerations are discussed in the main paper.

## C  DSGD with Optimal Client Sampling

### C.1  Proof of Theorem 13

*Proof.* $L$-smoothness of $f_i$ and the assumption on the gradient imply that the inequality

$$\mathrm{E}\left[\left\|g_i^k\right\|^2\right] \leq 2L(1+M)(f_i(x^k) - f_i(x^\star) + Z_i) + \sigma^2$$

holds for all $k \geq 0$. We first take expectations over $x^{k+1}$ conditioned on $x^k$ and over the sampling $S^k$:

$$\mathrm{E}\left[\left\|r^{k+1}\right\|^2\right] = \left\|r^k\right\|^2 - 2\eta^k \mathrm{E}\left[\left\langle \sum_{i \in S^k} \frac{w_i}{p_i^k} g_i^k, r^k \right\rangle\right] + (\eta^k)^2 \mathrm{E}\left[\left\|\sum_{i \in S^k} \frac{w_i}{p_i^k} g_i^k\right\|^2\right]$$

$$= \left\|r^k\right\|^2 - 2\eta^k \left\langle \nabla f(x^k), r^k \right\rangle + (\eta^k)^2 \left(\mathrm{E}\left[\left\|\sum_{i \in S^k} \frac{w_i}{p_i^k} g_i^k - \sum_{i=1}^{n} w_i g_i^k\right\|^2\right] + \mathrm{E}\left[\left\|\sum_{i=1}^{n} w_i g_i^k\right\|^2\right]\right)$$

$$\leq (1 - \mu\eta^k)\left\|r^k\right\|^2 - 2\eta^k \left(f(x^k) - f^\star\right) + (\eta^k)^2 \left(\mathrm{E}\left[\left\|\sum_{i \in S^k} \frac{w_i}{p_i^k} g_i^k - \sum_{i=1}^{n} w_i g_i^k\right\|^2\right] + \mathrm{E}\left[\left\|\sum_{i=1}^{n} w_i g_i^k\right\|^2\right]\right),$$

where

$$\mathrm{E}\left[\left\|\sum_{i \in S^k} \frac{w_i}{p_i^k} g_i^k - \sum_{i=1}^{n} w_i g_i^k\right\|^2\right] = \alpha^k \frac{n-m}{m} \mathrm{E}\left[\sum_{i=1}^{n} w_i^2 \left\|g_i^k\right\|^2\right]$$

$$= \alpha^k \frac{n-m}{m} \mathrm{E}\left[\sum_{i=1}^{n} w_i^2 \left(\left\|g_i^k - \nabla f_i(x^k)\right\|^2 + \left\|\nabla f_i(x^k)\right\|^2\right)\right]$$

$$= \alpha^k \frac{n-m}{m} \mathrm{E}\left[\sum_{i=1}^{n} w_i^2 \left(\left\|\xi_i^k\right\|^2 + \left\|\nabla f_i(x^k)\right\|^2\right)\right]$$

$$\leq \alpha^k \frac{n-m}{m} \sum_{i=1}^{n} w_i^2 \left(2L(1+M)(f_i(x^k) - f_i(x^\star) + Z_i) + \sigma^2\right)$$

$$\leq \alpha^k \frac{n-m}{m} \left(2WL(1+M)(f(x^k) - f^\star) + \sum_{i=1}^{n} w_i^2 (2L(1+M)Z_i + \sigma^2)\right),$$

and

$$
\begin{aligned}
\mathrm{E}\left[\left\|\sum_{i=1}^{n} w_i g_i^k\right\|^2\right] &= \mathrm{E}\left[\left\|\sum_{i=1}^{n} w_i g_i^k - \nabla f(x^k)\right\|^2\right] + \left\|\nabla f(x^k)\right\|^2 \\
&= \sum_{i=1}^{n} \mathrm{E}\left[\left\|w_i g_i^k - w_i \nabla f_i(x^k)\right\|^2\right] + \left\|\nabla f(x^k)\right\|^2 \\
&= \sum_{i=1}^{n} w_i^2 \mathrm{E}\left[\left\|\xi_i^k\right\|^2\right] + \left\|\nabla f(x^k)\right\|^2 \\
&\leq \sum_{i=1}^{n} w_i^2 (2LM(f_i(x^k) - f_i^\star) + \sigma^2) + 2L(f(x^k) - f^\star) \\
&= 2L(1 + WM)(f(x^k) - f^\star) + \sum_{i=1}^{n} w_i^2 (2LMZ_i + \sigma^2).
\end{aligned}
$$

Therefore, we obtain

$$
\begin{aligned}
\mathrm{E}\left[\left\|r^{k+1}\right\|^2\right] &\leq (1 - \mu\eta^k)\left\|r^k\right\|^2 - 2\eta^k\left(f(x^k) - f^\star\right) \\
&\quad + (\eta^k)^2 \left(2L(1 + WM)(f(x^k) - f^\star) + \sum_{i=1}^{n} w_i^2 (2LMZ_i + \sigma^2)\right) \\
&\quad + (\eta^k)^2 \alpha^k \frac{n-m}{m} \left(2WL(1+M)(f(x^k) - f^\star) + \sum_{i=1}^{n} w_i^2 (2L(1+M)Z_i + \sigma^2)\right) \\
&\leq (1 - \mu\eta^k)\left\|r^k\right\|^2 - 2\eta^k \left(1 - \eta^k \frac{(\alpha^k(n-m) + m)(1 + WM)L}{m}\right)\left(f(x^k) - f^\star\right) \\
&\quad + (\eta^k)^2 \frac{\alpha^k(n-m) + m}{m} \left(\sum_{i=1}^{n} w_i^2 (2L(1+M)Z_i + \sigma^2)\right) - (\eta^k)^2 2L \sum_{i=1}^{n} w_i^2 Z_i.
\end{aligned}
$$

Now choose any $0 < \eta^k \leq \frac{m}{(\alpha^k(n-m) + m)(1 + WM)L}$ and define

$$
\beta_1 := \sum_{i=1}^{n} w_i^2 (2L(1+M)Z_i + \sigma^2), \quad \beta_2 := 2L \sum_{i=1}^{n} w_i^2 Z_i, \quad \gamma^k := \frac{m}{\alpha^k(n-m) + m} \in \left[\frac{m}{n}, 1\right].
$$

Taking full expectation yields the desired result:

$$
\mathrm{E}\left[\left\|r^{k+1}\right\|^2\right] \leq (1 - \mu\eta^k)\mathrm{E}\left[\left\|r^k\right\|^2\right] + (\eta^k)^2 \left(\frac{\beta_1}{\gamma^k} - \beta_2\right).
$$

$\square$

## C.2   Proof of Theorem 15

*Proof.* Using equation (2), we have

$$
\begin{aligned}
f(x^{k+1}) &= f(x^k - \eta^k \mathbf{G}^k) \\
&= f(x^k) - \eta^k \left\langle \mathbf{G}^k, \nabla f(x^k)\right\rangle + \frac{(\eta^k)^2}{2}\left\langle \mathbf{G}^k, \nabla^2 f(z^k)\mathbf{G}^k\right\rangle, \quad \text{for some } z^k \in \mathbb{R}^d.
\end{aligned}
$$

Since all $f_i$'s are $L$-smooth, $f$ is also $L$-smooth. Therefore, we have $-L\boldsymbol{I} \preceq \nabla^2 f(x) \preceq L\boldsymbol{I}$ for all $x \in \mathbb{R}^d$. Combining this with the fact that $\mathbf{G}^k$ is an unbiased estimator of $\nabla f(x^k)$, we have

$$
\mathrm{E}\left[f(x^{k+1})\right] \leq f(x^k) - \eta^k \left\|\nabla f(x^k)\right\|^2 + \frac{(\eta^k)^2 L}{2}\mathrm{E}\left[\left\|\mathbf{G}^k\right\|^2\right], \tag{34}
$$

where the expectations are conditioned on $x^k$. In Appendix C.1, we already obtained the upper bound for the last term in equation (34):

$$\mathrm{E}\left[\left\|\mathbf{G}^k\right\|^2\right] \leq \left((1+M)\alpha^k\frac{n-m}{m} + M\right)\sum_{i=1}^{n} w_i^2 \left\|\nabla f_i(x^k)\right\|^2 + \left(\alpha^k\frac{n-m}{m} + 1\right)\sum_{i=1}^{n} w_i^2\sigma^2 + \left\|\nabla f(x^k)\right\|^2$$

$$= \left(\frac{1+M}{\gamma^k} - 1\right)\sum_{i=1}^{n} w_i^2 \left\|\nabla f_i(x^k)\right\|^2 + \frac{1}{\gamma^k}\sum_{i=1}^{n} w_i^2\sigma^2 + \left\|\nabla f(x^k)\right\|^2.$$

By Assumption 9, we further bound

$$\sum_{i=1}^{n} w_i^2 \left\|\nabla f_i(x^k)\right\|^2 \leq W\sum_{i=1}^{n} w_i \left\|\nabla f_i(x^k)\right\|^2$$

$$\leq W\left(\sum_{i=1}^{n} w_i \left\|\nabla f_i(x^k) - \nabla f(x^k)\right\|^2 + \left\|\nabla f(x^k)\right\|^2\right)$$

$$\leq W\rho + \left\|\nabla f(x^k)\right\|^2.$$

Combining the inequalities above and taking full expectation yields equation (24). $\qquad\square$

## D $\quad$ `FedAvg` **with Optimal Client Sampling**

**Lemma 21** ((Karimireddy et al., 2019))**.** *For any $L$-smooth and $\mu$-strongly convex function $h : \mathbb{R}^d \to \mathbb{R}$ and any $x, y, z \in \mathbb{R}^d$, the following inequality holds*

$$\langle\nabla h(x), z - y\rangle \geq h(z) - h(y) + \frac{\mu}{4}\left\|y - z\right\|^2 - L\left\|z - x\right\|^2. \tag{35}$$

*Proof.* For any given $x$, $y$, and $z$, the two inequalities below follows by the smoothness and strong convexity of the function $h$:

$$\langle\nabla h(x), z - x\rangle \geq h(z) - h(x) - \frac{L}{2}\left\|z - x\right\|^2,$$

$$\langle\nabla h(x), x - y\rangle \geq h(x) - h(y) + \frac{\mu}{2}\left\|y - x\right\|^2.$$

Further, applying the relaxed triangle inequality gives

$$\frac{\mu}{2}\left\|y - x\right\|^2 \geq \frac{\mu}{4}\left\|y - z\right\|^2 - \frac{\mu}{2}\left\|x - z\right\|^2.$$

Combining all these inequalities together we have

$$\langle\nabla h(x), z - y\rangle \geq h(z) - h(y) + \frac{\mu}{4}\left\|y - z\right\|^2 - \frac{L+\mu}{2}\left\|z - x\right\|^2.$$

The lemma follows by $L \geq \mu$. $\qquad\square$

### D.1 $\quad$ Proof of Theorem 17

*Proof.* The master update during round $k$ can be written as (superscript $k$ is dropped from here onward)

$$\eta_g\Delta x = \frac{\eta}{R}\sum_{i\in S,r}\frac{w_i}{p_i}g_i(y_{i,r-1}) \quad \text{and} \quad \mathrm{E}\left[\eta_g\Delta x\right] = \frac{\eta}{R}\sum_{i,r} w_i\mathrm{E}\left[\nabla f_i(y_{i,r-1})\right].$$

Summations are always over $i \in [n]$ and $r \in [R]$ unless stated otherwise. Taking expectations over $x$ conditioned on the results prior to round $k$ and over the sampling $S$ gives

$$
\mathrm{E}\left[\|x - \eta_g \Delta x - x^\star\|^2\right] = \|x - x^\star\|^2 \underbrace{- \frac{2\eta}{R} \sum_{i,r} \langle w_i \nabla f_i(y_{i,r-1}), x - x^\star \rangle}_{\mathcal{A}_1} + \underbrace{\frac{\eta^2}{R^2} \mathrm{E}\left[\left\|\sum_{i \in S,r} \frac{w_i}{p_i} g_i(y_{i,r-1})\right\|^2\right]}_{\mathcal{A}_2}.
$$

Applying Lemma 21 with $h = w_i f_i$, $x = y_{i,r-1}$, $y = x^\star$ and $z = x$ gives

$$
\mathcal{A}_1 \leq -\frac{2\eta}{R} \sum_{i,r} \left(w_i f_i(x) - w_i f_i(x^\star) + w_i \frac{\mu}{4} \|x - x^\star\|^2 - w_i L \|x - y_{i,r-1}\|^2\right)
$$

$$
\leq -2\eta \left(f(x) - f^\star + \frac{\mu}{4} \|x - x^\star\|^2\right) + 2L\eta\mathcal{E},
$$

where $\mathcal{E}$ is the drift caused by the local updates on the clients:

$$
\mathcal{E} := \frac{1}{R} \sum_{i,r} w_i \mathrm{E}\left[\|x - y_{i,r-1}\|^2\right]. \tag{36}
$$

Bounding $\mathcal{A}_2$, we obtain

$$
\frac{1}{\eta^2} \mathcal{A}_2 = \mathrm{E}\left[\left\|\sum_{i \in S} \frac{w_i}{p_i} \frac{1}{R} \sum_r g_i(y_{i,r-1}) - \sum_i w_i \frac{1}{R} \sum_r g_i(y_{i,r-1})\right\|^2\right] + \mathrm{E}\left[\left\|\sum_i w_i \frac{1}{R} \sum_r g_i(y_{i,r-1})\right\|^2\right]
$$

$$
\leq \alpha \frac{n-m}{m} \sum_i w_i^2 \mathrm{E}\left[\left\|\frac{1}{R} \sum_r g_i(y_{i,r-1})\right\|^2\right] + \mathrm{E}\left[\left\|\sum_i w_i \frac{1}{R} \sum_r g_i(y_{i,r-1})\right\|^2\right]
$$

$$
= \alpha \frac{n-m}{m} \sum_i w_i^2 \left(\mathrm{E}\left[\left\|\frac{1}{R} \sum_r \xi_{i,r-1}\right\|^2\right] + \mathrm{E}\left[\left\|\frac{1}{R} \sum_r \nabla f_i(y_{i,r-1})\right\|^2\right]\right)
$$

$$
+ \mathrm{E}\left[\left\|\sum_i w_i \frac{1}{R} \sum_r \xi_{i,r-1}\right\|^2\right] + \mathrm{E}\left[\left\|\sum_i w_i \frac{1}{R} \sum_r \nabla f_i(y_{i,r-1})\right\|^2\right].
$$

Using independence, zero mean and bounded second moment of the random variables $\xi_{i,r}$, we obtain

$$
\begin{aligned}
\frac{1}{\eta^2}\mathcal{A}_2 &\le \alpha\frac{n-m}{m}\sum_i w_i^2\left(\frac{1}{R^2}\sum_r \mathrm{E}\left[\|\xi_{i,r-1}\|^2\right] + \mathrm{E}\left[\left\|\frac{1}{R}\sum_r \nabla f_i(y_{i,r-1})\right\|^2\right]\right)\\
&\quad + \sum_i w_i^2\frac{1}{R^2}\sum_r \mathrm{E}\left[\|\xi_{i,r-1}\|^2\right] + \mathrm{E}\left[\left\|\sum_i w_i\frac{1}{R}\sum_r \nabla f_i(y_{i,r-1})\right\|^2\right]\\
&\le \alpha\frac{n-m}{m}\sum_i w_i^2\left(\left(\frac{M}{R^2}+\frac{1}{R}\right)\sum_r \mathrm{E}\left[\|\nabla f_i(y_{i,r-1})\|^2\right] + \frac{\sigma^2}{R}\right)\\
&\quad + \sum_i w_i^2\left(\frac{M}{R^2}\sum_r \mathrm{E}\left[\|\nabla f_i(y_{i,r-1})\|^2\right] + \frac{\sigma^2}{R}\right) + \mathrm{E}\left[\left\|\sum_i w_i\frac{1}{R}\sum_r \nabla f_i(y_{i,r-1})\right\|^2\right]\\
&= \frac{\sigma^2}{R\gamma}\sum_i w_i^2 + \left(\frac{M}{R}+\left(\frac{M}{R}+1\right)\alpha\frac{n-m}{m}\right)\sum_i w_i^2\frac{1}{R}\sum_r \mathrm{E}\left[\|\nabla f_i(y_{i,r-1})-\nabla f_i(x)+\nabla f_i(x)\|^2\right]\\
&\quad + \mathrm{E}\left[\left\|\sum_i w_i\frac{1}{R}\sum_r (\nabla f_i(y_{i,r-1})-\nabla f_i(x))+\nabla f(x)\right\|^2\right]\\
&\le \frac{\sigma^2}{R\gamma}\sum_i w_i^2 + \left(\frac{M}{R}+\left(\frac{M}{R}+1\right)\alpha\frac{n-m}{m}\right)\sum_i w_i^2\left(\frac{2}{R}\sum_r \mathrm{E}\left[\|\nabla f_i(y_{i,r-1})-\nabla f_i(x)\|^2\right] + 2\mathrm{E}\left[\|\nabla f_i(x)\|^2\right]\right)\\
&\quad + 2\mathrm{E}\left[\left\|\sum_i w_i\frac{1}{R}\sum_r (\nabla f_i(y_{i,r-1})-\nabla f_i(x))\right\|^2\right] + 2\mathrm{E}\left[\|\nabla f(x)\|^2\right].
\end{aligned}
$$

Combining the smoothness of $f_i$'s, the definition of $\mathcal{E}$, and Jensen's inequality with definition $\gamma := \frac{m}{\alpha(n-m)+m}$, we obtain

$$
\begin{aligned}
\frac{1}{\eta^2}\mathcal{A}_2 &\le \frac{\sigma^2}{R\gamma}\sum_i w_i^2 + 2\left(\frac{M}{R}+\left(\frac{M}{R}+1\right)\alpha\frac{n-m}{m}\right)\left(WL^2\mathcal{E}+2WL(f(x)-f^\star)+2L\sum_i w_i^2 Z_i\right)\\
&\quad + 2L^2\mathcal{E} + 4L(f(x)-f(x^\star))\\
&= \frac{\sigma^2}{R\gamma}\sum_i w_i^2 + 2L^2\left((1-W)+\frac{W}{\gamma}\left(\frac{M}{R}+1\right)\right)\mathcal{E} + 4L\left(\frac{1}{\gamma}\left(\frac{M}{R}+1\right)-1\right)\sum_i w_i^2 Z_i\\
&\quad + 4L\left((1-W)+\frac{W}{\gamma}\left(\frac{M}{R}+1\right)\right)(f(x)-f^\star).
\end{aligned}
$$

Putting these bounds on $\mathcal{A}_1$ and $\mathcal{A}_2$ together and using the fact that $1-W \le 1/\gamma$ yields

$$
\begin{aligned}
\mathrm{E}\left[\|x-\eta_g\Delta x-x^\star\|^2\right] &\le \left(1-\frac{\mu\eta}{2}\right)\|x-x^\star\|^2 - 2\eta\left(1-2L\frac{\eta}{\gamma}\left(W\left(\frac{M}{R}+1\right)+1\right)\right)(f(x)-f^\star)\\
&\quad + \eta^2\left(\frac{\sigma^2}{R\gamma}\sum_i w_i^2 + 4L\left(\frac{1}{\gamma}\left(\frac{M}{R}+1\right)-1\right)\sum_i w_i^2 Z_i\right)\\
&\quad + \left(1+\eta L\left((1-W)+\frac{W}{\gamma}\left(\frac{M}{R}+1\right)\right)\right)2L\eta\mathcal{E}.
\end{aligned}
$$

Let $\eta \le \frac{\gamma}{8(1+W(1+M/R))L}$, then

$$
\frac{3}{4}\le 1-2L\frac{\eta}{\gamma}\left(W\left(\frac{M}{R}+1\right)+1\right),
$$

which in turn yields

$$
\begin{aligned}
\mathrm{E}\left[\|x - \eta_g \Delta x - x^\star\|^2\right] \leq{}& \left(1 - \frac{\mu\eta}{2}\right)\|x - x^\star\|^2 - \frac{3\eta}{2}(f(x) - f^\star) \\
&+ \eta^2\left(\frac{\sigma^2}{R\gamma}\sum_i w_i^2 + 4L\left(\frac{1}{\gamma}\left(\frac{M}{R}+1\right)-1\right)\sum_i w_i^2 Z_i\right) \\
&+ \left(1 + \eta L\left((1-W) + \frac{W}{\gamma}\left(\frac{M}{R}+1\right)\right)\right)2L\eta\mathcal{E}.
\end{aligned}
\tag{37}
$$

Next, we need to bound the drift $\mathcal{E}$. For $R \geq 2$, we have

$$
\begin{aligned}
\mathrm{E}\left[\|y_{i,r} - x\|^2\right] &= \mathrm{E}\left[\|y_{i,r-1} - x - \eta_l g_i(y_{i,r-1})\|^2\right] \\
&\leq \mathrm{E}\left[\|y_{i,r-1} - x - \eta_l \nabla f_i(y_{i,r-1})\|^2\right] + \eta_l^2(M\|\nabla f_i(y_{i,r-1})\|^2 + \sigma^2) \\
&\leq \left(1 + \frac{1}{R-1}\right)\mathrm{E}\left[\|y_{i,r-1} - x\|^2\right] + (R+M)\eta_l^2\|\nabla f_i(y_{i,r-1})\|^2 + \eta_l^2\sigma^2 \\
&= \left(1 + \frac{1}{R-1}\right)\mathrm{E}\left[\|y_{i,r-1} - x\|^2\right] + \left(1 + \frac{M}{R}\right)\frac{\eta^2}{R\eta_g^2}\|\nabla f_i(y_{i,r-1})\|^2 + \frac{\eta^2\sigma^2}{R^2\eta_g^2} \\
&\leq \left(1 + \frac{1}{R-1}\right)\mathrm{E}\left[\|y_{i,r-1} - x\|^2\right] + \left(1 + \frac{M}{R}\right)\frac{2\eta^2}{R\eta_g^2}\|\nabla f_i(y_{i,r-1}) - \nabla f_i(x)\|^2 \\
&\quad + \left(1 + \frac{M}{R}\right)\frac{2\eta^2}{R\eta_g^2}\|\nabla f_i(x)\|^2 + \frac{\eta^2\sigma^2}{R^2\eta_g^2} \\
&\leq \left(1 + \frac{1}{R-1} + \left(1 + \frac{M}{R}\right)\frac{2\eta^2 L^2}{R\eta_g^2}\right)\mathrm{E}\left[\|y_{i,r-1} - x\|^2\right] + \left(1 + \frac{M}{R}\right)\frac{2\eta^2}{R\eta_g^2}\|\nabla f_i(x)\|^2 + \frac{\eta^2\sigma^2}{R^2\eta_g^2}.
\end{aligned}
$$

If we further restrict $\eta \leq \frac{1}{8L(2+M/R)}$, then for any $\eta_g \geq 1$, we have

$$
\left(1 + \frac{M}{R}\right)\frac{2\eta^2 L^2}{R\eta_g^2} \leq \frac{2L^2}{R\eta_g^2}\frac{1}{64L^2} \leq \frac{1}{32R} \leq \frac{1}{32(R-1)},
$$

and therefore,

$$
\begin{aligned}
\mathrm{E}\left[\|y_{i,r} - x\|^2\right] &\leq \left(1 + \frac{33}{32(R-1)}\right)\mathrm{E}\left[\|y_{i,r-1} - x\|^2\right] + \left(1 + \frac{M}{R}\right)\frac{2\eta^2}{R\eta_g^2}\|\nabla f_i(x)\|^2 + \frac{\eta^2\sigma^2}{R^2\eta_g^2} \\
&\leq \sum_{\tau=0}^{r-1}\left(1 + \frac{33}{32(R-1)}\right)^\tau\left(\left(1 + \frac{M}{R}\right)\frac{2\eta^2}{R\eta_g^2}\|\nabla f_i(x)\|^2 + \frac{\eta^2\sigma^2}{R^2\eta_g^2}\right) \\
&\leq 8R\left(\left(1 + \frac{M}{R}\right)\frac{2\eta^2}{R\eta_g^2}\|\nabla f_i(x)\|^2 + \frac{\eta^2\sigma^2}{R^2\eta_g^2}\right) \\
&= 16\left(1 + \frac{M}{R}\right)\eta^2\|\nabla f_i(x)\|^2 + \frac{8\eta^2\sigma^2}{R\eta_g^2}.
\end{aligned}
$$

Hence, the drift is bounded by

$$
\begin{aligned}
\mathcal{E} &\leq 16\left(1+\frac{M}{R}\right)\eta^2\sum_i w_i\|\nabla f_i(x)\|^2 + \frac{8\eta^2\sigma^2}{R\eta_g^2}\\
&\leq 32\left(1+\frac{M}{R}\right)\eta^2 L\sum_i w_i(f_i(x)-f_i^\star) + \frac{8\eta^2\sigma^2}{R\eta_g^2}\\
&= 32\left(1+\frac{M}{R}\right)\eta^2 L(f(x)-f^\star) + 32\left(1+\frac{M}{R}\right)\eta^2 L\sum_i w_i Z_i + \frac{8\eta^2\sigma^2}{R\eta_g^2}\\
&\leq 4\eta(f(x)-f^\star) + 32\left(1+\frac{M}{R}\right)\eta^2 L\sum_i w_i Z_i + \frac{8\eta^2\sigma^2}{R\eta_g^2}.
\end{aligned}
$$

Due to the upper bound on the step size $\eta \leq \frac{1}{8L(2+M/R)}$, we have the inequalities

$$
1+\eta L\left((1-W)+\frac{W}{\gamma}\left(\frac{M}{R}+1\right)\right) \leq \frac{9}{8} \quad\text{and}\quad 8\eta L \leq 1. \tag{38}
$$

Plugging these to (37), we obtain

$$
\begin{aligned}
\mathrm{E}\left[\|x-\eta_g\Delta x-x^\star\|^2\right] &\leq \left(1-\frac{\mu\eta}{2}\right)\|x-x^\star\|^2 - \frac{3}{8}\eta(f(x)-f^\star)\\
&\quad + \eta^2\left(\frac{\sigma^2}{\gamma R}\left(\frac{\gamma}{\eta_g^2}+\sum_i w_i^2\right)+4L\left(\frac{M}{R}+1-\gamma\right)\sum_i w_i^2 Z_i\right)\\
&\quad + \eta^3 72L^2\left(1+\frac{M}{R}\right)\sum_i w_i Z_i.
\end{aligned}
$$

Rearranging the terms in the last inequality, taking full expectation and including superscripts lead to

$$
\begin{aligned}
\frac{3}{8}\mathrm{E}\left[(f(x^k)-f^\star)\right] &\leq \frac{1}{\eta^k}\left(1-\frac{\mu\eta^k}{2}\right)\mathrm{E}\left[\|x^k-x^\star\|^2\right] - \frac{1}{\eta^k}\mathrm{E}\left[\|x^{k+1}-x^\star\|^2\right]\\
&\quad + \eta^k\left(\frac{\sigma^2}{\gamma^k R}\left(\frac{\gamma^k}{\eta_g^2}+\sum_i w_i^2\right)+4L\left(\frac{M}{R}+1-\gamma^k\right)\sum_i w_i^2 Z_i\right)\\
&\quad + (\eta^k)^2 72L^2\left(1+\frac{M}{R}\right)\sum_i w_i Z_i.
\end{aligned}
$$

Plugging the assumption $\eta_g^k \geq \sqrt{\frac{\gamma^k}{\sum_i w_i^2}}$ into the RHS of the above inequality completes the proof.

$\square$

## D.2 Proof of Theorem 18

*Proof.* We drop superscript $k$ and write the master update during round $k$ as:

$$
\eta_g\Delta x = \frac{\eta}{R}\sum_{i\in S,r}\frac{w_i}{p_i}g_i(y_{i,r-1}) := \eta\tilde{\Delta}.
$$

Summations are always over $i \in [n]$ and $r \in [R]$ unless stated otherwise. Taking expectations conditioned on $x$ and using a similar argument as in the proof in Appendix C.2, we have

$$\mathrm{E}\left[f(x - \eta_g \Delta x)\right] \leq f(x) - \eta \left\langle \nabla f(x), \mathrm{E}\left[\tilde{\Delta}\right] \right\rangle + \frac{\eta^2 L}{2} \mathrm{E}\left[\left\|\tilde{\Delta}\right\|^2\right]$$

$$= f(x) - \eta \left\|\nabla f(x)\right\|^2 + \eta \left\langle \nabla f(x), \nabla f(x) - \mathrm{E}\left[\tilde{\Delta}\right] \right\rangle + \frac{\eta^2 L}{2} \mathrm{E}\left[\left\|\tilde{\Delta}\right\|^2\right]$$

$$\leq f(x) - \frac{\eta}{2} \left\|\nabla f(x)\right\|^2 + \frac{\eta}{2} \mathrm{E}\left[\left\|\nabla f(x) - \mathrm{E}_S\left[\tilde{\Delta}\right]\right\|^2\right] + \frac{\eta^2 L}{2} \mathrm{E}\left[\left\|\tilde{\Delta}\right\|^2\right],$$

where the last inequality follows since $\langle a, b \rangle \leq \frac{1}{2}\|a\|^2 + \frac{1}{2}\|b\|^2$, $\forall a, b \in \mathbb{R}^d$. Since $f_i$'s are $L$-smooth, by the (relaxed) triangular inequality, we have

$$\frac{\eta}{2} \mathrm{E}\left[\left\|\nabla f(x) - \mathrm{E}\left[\tilde{\Delta}\right]\right\|^2\right] = \frac{\eta}{2} \mathrm{E}\left[\left\|\frac{1}{R} \sum_{i,r} w_i \left(\nabla f_i(x) - \nabla f_i(y_{i,r-1})\right)\right\|^2\right]$$

$$\leq \frac{\eta L^2}{2R} \sum_{i,r} w_i \mathrm{E}\left[\left\|x - y_{i,r-1}\right\|^2\right] = \frac{\eta L^2}{2} \mathcal{E},$$

where $\mathcal{E}$ is the drift caused by the local updates on the clients as defined in (36).

In Appendix D.1, we already obtained the upper bound for $\frac{1}{\eta^2} \mathcal{A}_2 = \mathrm{E}\left[\left\|\tilde{\Delta}\right\|^2\right]$:

$$\mathrm{E}\left[\left\|\tilde{\Delta}\right\|^2\right] \leq \frac{\sigma^2}{R\gamma} \sum_i w_i^2 + 2W\left(\frac{M}{R} + \left(\frac{M}{R} + 1\right)\alpha \frac{n-m}{m}\right)\left(L^2\mathcal{E} + \sum_i w_i \left\|\nabla f_i(x)\right\|^2\right) + 2L^2\mathcal{E} + 2\left\|\nabla f(x)\right\|^2.$$

Together with Assumption 9 that

$$\sum_i w_i \left\|\nabla f_i(x)\right\|^2 - \left\|\nabla f(x)\right\|^2 \leq \sum_i w_i \left\|\nabla f_i(x) - \nabla f(x)\right\|^2 \leq \rho,$$

we have

$$\mathrm{E}\left[\left\|\tilde{\Delta}\right\|^2\right] \leq \frac{\sigma^2}{R\gamma} \sum_i w_i^2 + \frac{2W}{\gamma}\left(\frac{M}{R} + 1 - \gamma\right)\left(L^2\mathcal{E} + \left\|\nabla f(x)\right\|^2 + \rho\right) + 2L^2\mathcal{E} + 2\left\|\nabla f(x)\right\|^2.$$

Combining the above inequalities gives

$$\mathrm{E}\left[f(x - \eta_g \Delta x)\right] \leq f(x) + \eta^2 \frac{\sigma^2 L}{2R\gamma} \sum_i w_i^2 + \eta L^2\left(\eta L\left((1-W) + \frac{W}{\gamma}\left(1 + \frac{M}{R}\right)\right) + \frac{1}{2}\right)\mathcal{E}$$

$$+ \eta\left(\eta L\left((1-W) + \frac{W}{\gamma}\left(1 + \frac{M}{R}\right)\right) - \frac{1}{2}\right)\left\|\nabla f(x)\right\|^2$$

$$+ \eta\left(\eta L\left((1-W) + \frac{W}{\gamma}\left(1 + \frac{M}{R}\right)\right) - \eta L\right)\rho.$$

Now, applying inequality (38) gives

$$\mathrm{E}\left[f(x - \eta_g \Delta x)\right] \leq f(x) + \frac{\eta^2 \sigma^2 L}{2R\gamma} \sum_i w_i^2 + \frac{5\eta L^2}{8}\mathcal{E} - \frac{3\eta}{8}\left\|\nabla f(x)\right\|^2 + \frac{\eta}{8}(1 - 8\eta L)\rho.$$

In Appendix D.1, we also obtained the upper bound for the drift $\mathcal{E}$:

$$\mathcal{E} \leq 16\left(1 + \frac{M}{R}\right)\eta^2 \sum_i w_i \left\|\nabla f_i(x)\right\|^2 + \frac{8\eta^2 \sigma^2}{R\eta_g^2}$$

$$\leq 16\left(1 + \frac{M}{R}\right)\eta^2(\left\|\nabla f(x)\right\|^2 + \rho) + \frac{8\eta^2 \sigma^2}{R\eta_g^2}.$$

Since $8\eta L \leq 8\eta L(1 + {}^M/_R) \leq 1$, we have

$$\frac{5\eta L^2}{8}\mathcal{E} \leq 10\eta^3 L^2 \left(1 + \frac{M}{R}\right)(\|\nabla f(x)\|^2 + \rho) + \frac{5\eta^3 L^2 \sigma^2}{R\eta_g^2}$$

$$\leq \frac{5\eta^2 L}{4}(\|\nabla f(x)\|^2 + \rho) + \frac{5\eta^2 L\sigma^2}{8R\eta_g^2}.$$

This further simplifies the iterate to

$$\mathrm{E}\left[f(x - \eta_g \Delta x)\right] \leq f(x) - \frac{3}{8}\eta\left(1 - \frac{10}{3}\eta L\right)\|\nabla f(x)\|^2 + \frac{1}{8}\eta\left(1 + 2\eta L\right)\rho + \frac{\eta^2 \sigma^2 L}{2R\gamma}\left(\frac{5\gamma}{4\eta_g^2} + \sum_i w_i^2\right).$$

Applying the assumption that $\eta_g \geq \sqrt{\frac{5\gamma}{4\sum_i w_i^2}}$ and taking full expectations completes the proof:

$$\mathrm{E}\left[f(x - \eta_g \Delta x)\right] \leq \mathrm{E}\left[f(x)\right] - \frac{3}{8}\eta\left(1 - \frac{10}{3}\eta L\right)\mathrm{E}\left[\|\nabla f(x)\|^2\right] + \eta\frac{\rho}{8} + \eta^2\left(\frac{\rho}{4} + \frac{\sigma^2}{R\gamma}\sum_{i=1}^n w_i^2\right)L.$$

$\square$

## E   A Sketch of Results on Partial Participation

This section discusses how our analysis can be extended to the case where not all clients are available to participate in each round. As an illustrative example, we consider Distributed SGD (DSGD), i.e., $\mathbf{U}_i^k = g_i^k$.

If not all clients are available to participate in each communication round, we will assume that there is a known distribution of client availability $\mathcal{Q}$ such that in each step a subset $\mathcal{Q}^k \sim \mathcal{Q}$ of clients are available to participate in a given communication round $k$. We denote the probability that client $i$ is available in the current run by $q_i$, i.e., $q_i = \mathrm{Prob}(i \in \mathcal{Q}^k)$. Under this setting, we can apply twice tower property of the expectation and obtain the following variance decomposition:

$$\mathrm{E}\left[\left\|\mathbf{G}^k - \nabla f(x^k)\right\|^2\right]$$

$$= \mathrm{E}\left[\mathrm{E}\left[\left\|\mathbf{G}^k - \sum_{i \in \mathcal{Q}^k}\frac{w_i}{q_i}\mathbf{U}_i^k\right\|^2 \bigg| \mathcal{Q}^k\right]\right] + \mathrm{E}\left[\left\|\sum_{i \in \mathcal{Q}^k}\frac{w_i}{q_i}\mathbf{U}_i^k - \sum_{i=1}^n w_i\mathbf{U}_i^k\right\|^2\right] + \mathrm{E}\left[\left\|\sum_{i=1}^n w_i\mathbf{U}_i^k - \nabla f(x^k)\right\|^2\right],$$
(39)

where we update the definition of $\mathbf{G}^k$

$$\mathbf{G}^k := \sum_{i \in S^k \subseteq \mathcal{Q}^k}\frac{w_i}{q_i p_i^k}.$$
(40)

Note that $S^k \subseteq \mathcal{Q}^k$ as we can only sample from available clients. Furthermore, in the particular case where all clients are available, the above equations become identical to the ones that we present in the main paper.

Upper-bounding Equation (39) in an analogous way as we proceed in our analysis in Appendices C and D would complete the proof of convergence for these settings.

## F   Experimental Details

### F.1   Federated EMNIST Dataset

We detail the hyper-parameters used in the experiments on the FEMNIST datasets. For each experiment, we run 151 communication rounds, reporting (local) training loss every round and validation accuracy every

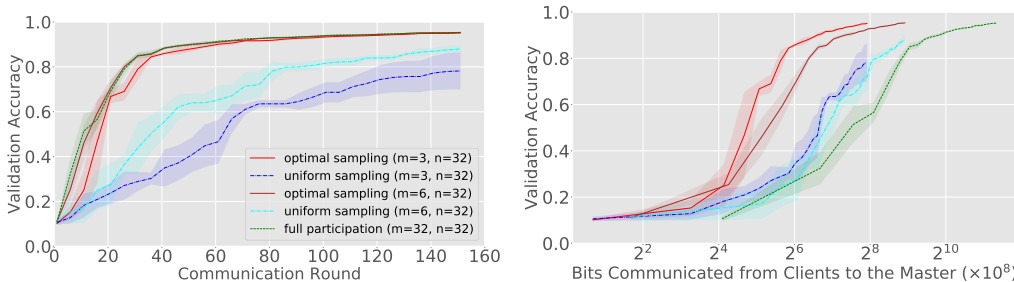

Figure 8: (FEMNIST Dataset 1, $n = 32$) current best validation accuracy as a function of the number of communication rounds and the number of bits communicated from clients to the master.

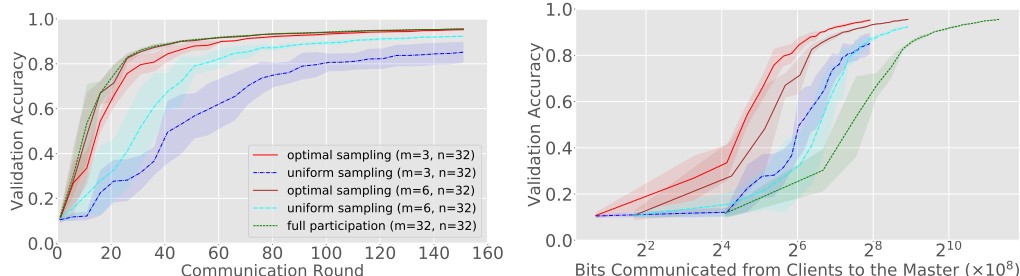

Figure 9: (FEMNIST Dataset 2, $n = 32$) current best validation accuracy as a function of the number of communication rounds and the number of bits communicated from clients to the master.

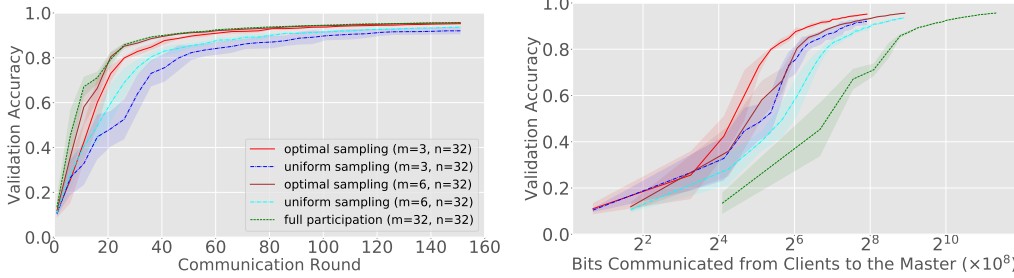

Figure 10: (FEMNIST Dataset 3, $n = 32$) current best validation accuracy as a function of the number of communication rounds and the number of bits communicated from clients to the master.

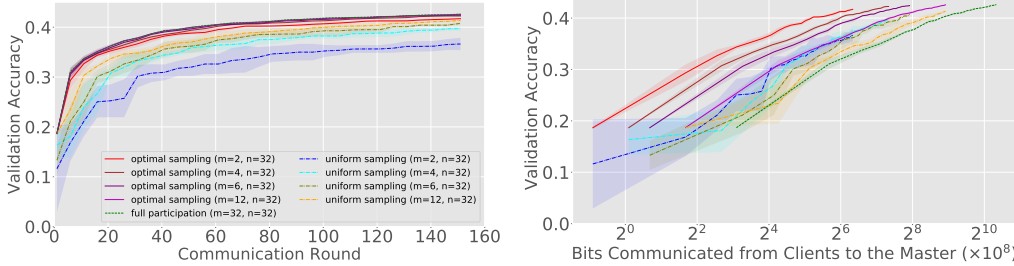

Figure 11: (Shakespeare Dataset, $n = 32$) current best validation accuracy as a function of the number of communication rounds and the number of bits communicated from clients to the master.

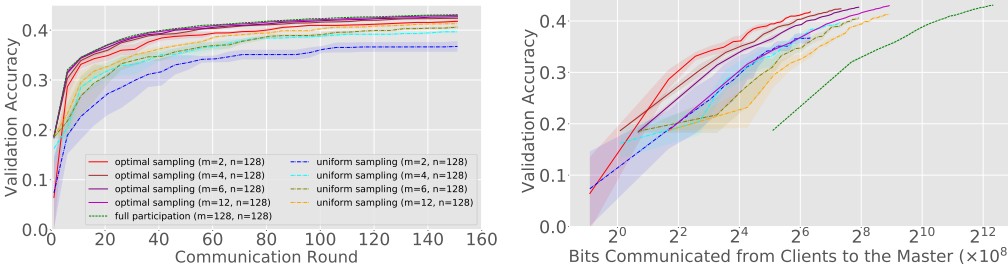

Figure 12: (Shakespeare Dataset, $n = 128$) current best validation accuracy as a function of the number of communication rounds and the number of bits communicated from clients to the master.

5 rounds. In each round, $n = 32$ clients are sampled from the client pool, each of which then performs SGD for 1 epoch on its local training images with batch size 20. For partial participation, the expected number of clients allowed to communicate their updates back to the master is set to $m \in \{3, 6\}$. We use vanilla SGD and constant step sizes for all experiments, where we set $\eta_g = 1$ and tune $\eta_l$ from the set of value $\{2^{-1}, 2^{-2}, 2^{-3}, 2^{-4}, 2^{-5}\}$. If the optimal step size hits a boundary value, then we try one more step size by extending that boundary and repeat this until the optimal step size is not a boundary value. For full participation and optimal sampling, it turns out that $\eta_l = 2^{-3}$ is the optimal local step size for all three datasets. For uniform sampling, the optimal is $\eta_l = 2^{-5}$ for Dataset 1 and $\eta_l = 2^{-4}$ for Datasets 2 and 3. For the extra communications in Algorithm 2, we set $j_{max} = 4$.

We also present some additional figures of the experiment results. Figures 8, 9 and 10 show the current best validation accuracy as a function of the number of communication rounds and the number of bits communicated from clients to the master on Datasets 1, 2 and 3, respectively.

### F.2 Shakespeare Dataset

We detail the hyper-parameters used in the experiments on the Shakespeare dataset. For each experiment, we run 151 communication rounds, reporting (local) training loss every round and validation accuracy every 5 rounds. In each round, $n \in \{32, 128\}$ clients are sampled from the client pool, each of which then performs SGD for 1 epoch on its local training data with batch size 8 (each batch contains 8 example sequences of length 5). For partial participation, the expected number of clients allowed to communicate their updates back to the master is set to $m \in \{2, 4, 6, 12\}$. We use vanilla SGD and constant step sizes for all experiments, where we set $\eta_g = 1$ and tune $\eta_l$ from the set of value $\{2^{-1}, 2^{-2}, 2^{-3}, 2^{-4}, 2^{-5}\}$. If the optimal step size hits a boundary value, then we try one more step size by extending that boundary and repeat this until the optimal step size is not a boundary value. For full participation and optimal sampling, it turns out that $\eta_l = 2^{-2}$ is the optimal local step size. For uniform sampling, the optimal is $\eta_l = 2^{-3}$. For the extra communications in Algorithm 2, we set $j_{max} = 4$.

We also present an additional figure of the experiment result. Figures 11 and 12 show the current best validation accuracy as a function of the number of communication rounds and the number of bits communicated from clients to the maste for the cases $n = 32, 128$, respectively.

## G  Additional Experiment on Federated CIFAR100 Dataset

We evaluate our method on the Federated CIFAR100 image dataset for image classification. The Federated CIFAR100 dataset is a balanced dataset, where every client holds the same number of training images. In each communication round, $n = 32$ clients are sampled uniformly from the client pool, each of which then performs several SGD steps on its local training images for 1 epoch with batch size 20. This means that all clients have the same number of local steps in each round. For partial participation, the expected number of clients allowed to communicate their updates back to the master is set to $m = 3$. We use vanilla SGD optimizers with constant step sizes for both clients and the master, with $\eta_g = 1$ and $\eta_l$ tuned on a holdout

Figure 13: (CIFAR100 Dataset, $n = 32$) Validation accuracy and (local) training loss as a function of the number of communication rounds and the number of bits communicated from clients to the master.

set. For full participation and optimal sampling, it turns out that $\eta_l = 1 \times 10^{-3}$ is the optimal local step size. For uniform sampling, the optimal is $\eta_l = 3 \times 10^{-4}$. We set $j_{\max} = 4$ and include the extra communication costs in our results. The main results are shown in Figure 13. It can be seen that our optimal client sampling scheme achieves better performance than uniform sampling on this balanced dataset. The performance gains of our method over uniform sampling come from the fact that the norms of the updates from some clients are larger than those from other clients even if all clients run the same number of local steps in each round.

