# OpenReview forum: "Optimal Client Sampling for Federated Learning"
_TMLR — Accepted by TMLR_

### Review · Reviewer_w6Qr · 2022-06-15

**Summary Of Contributions:**

This paper analyzes strategies for reducing the amount of client-to-server communication in federated learning (FL) via client sampling. This work gives a client sampling mechanism that is optimal (in terms of its variance). Since this optimal mechanism is not always compatible with the secure aggregation protocol (SecAgg), the authors also give a variant that employs iterative clipping that is compatible with SecAgg.

The authors then perform convergence analyses of Federated Averaging (FedAvg) and Distributed SGD (DSGD) when using the optimal client sampling mechanism. Finally, the authors give empirical comparisons of FedAvg on two primary tasks. Notably, these empirical comparisons show that the optimal client sampling mechanism achieves better accuracy for less communication, while maintaining accuracy comparable to that of not employing any sampling mechanism.

**Broader Impact Concerns:**

I have no concerns in this regard.

**Requested Changes:**

## Critical Changes (Necessary for Acceptance)

### Adjusting Discussion of Contributions in Section 2

As discussed above, I believe that the contributions of Section 2 must be rephrased, especially to reference past works [1, 2] above. In particular, avoiding reference to the optimal sampling mechanism being proposed in this work. While I'd like to leave the authors with some leeway here, I believe that this section needs to be reworked in a way that differentiates it from other works, either through a broader exploration of prior work on sampling and why various methods are useful/not useful for FL, or some kind of improved analysis (eg. a variance analysis of the iterative thresholding in Algorithm 2).

### More Nuanced Discussion of Privacy

The authors mention that their scheme (particularly Algorithm 2) does not compromise client privacy, and that they "fulfill the privacy requirements of FL." This is a bit too broad of a characterization of privacy in FL, and could benefit from more nuance. In particular, SecAgg-compatibility is only one form of privacy. Differential privacy (DP) is yet another. While I don't believe this paper needs to consider DP, the contributions of this work should be clarified to say that the authors propose algorithms that are compatible with SecAgg. In general, the privacy requirements of FL are a fluid, application-dependent thing, and in some cases SecAgg-compatibility is not enough to be considered privacy-preserving. In this vein, please consider changing the title of Section 2.2 to something like "Ensuring Compatibility with Secure Aggregation" to be more clear.

### Improving Experimental Reproducibility

I would request that the authors detail how Datasets 1, 2, and 3 were created from FEMNIST in the paper itself. While the datasets are available through a link in a README in the supplement, the method for creating these splits could be important for related types of research in the future.

## Recommended Changes

### Sketching Results on Partial Participation

Footnote 1 is an important note to consider, as the settings where client aggregation is a bottleneck are also often settings where there is only partial participation of clients. While I am convinced that the results in the manuscript can be extended to this setting, I think it would be beneficial to have a limited sketch of this in the appendix for the interested reader. Notably, while the extension of the client sampling scheme to this setting is effectively trivial, incorporation into the actual convergence rates is slightly less trivial and deserves (in my opinion) some elucidation.

### Avoiding Brevity to Improve Readability

The work could benefit from being slightly more verbose than in its current phase, especially given the lack of a page limit in TMLR. For example, the "Interpretation" paragraphs in Section 3 are extremely useful, and could be expanded upon to improve readability. For instance, in the paragraph after Theorem 3.6, the authors refer to specific specializations of equation 17. I believe that this would be improved by actually plugging in the specializations and referencing specific equations in past works that mirror these (eg. what theorem in Gower et al., 2019 is being referenced. Similarly, I believe that the interpretation section after Theorem 3.9 could be significantly expanded. For example, Theorem 3.8 involves somewhat complicated $\beta_i$ terms that could be discussed at least at a high level.

Related to the above (though perhaps more specific) would be brief sketches of the proofs of theorems in Section 3. In particular, given the wide array of existing analyses of DSGD and FedAvg, trying to disambiguate why the arguments are more than simply plugging in the variance estimates from Section 2.2 to existing results would be useful.

This same issue manifests in Section 4, which could use a bit more explanation. For example, it is not immediately clear based only on this work why the sampling schemes of Cho et al., 2020 are not compatible with SecAgg. For example, one way to bias towards clients with higher losses would be to sum all the client losses (which is SecAgg compatible) send the result back to the clients, and have clients report back with probability proportional to the ratio of their loss to the total loss. Maybe a better way to approach this would be to outline the selection strategy in Cho et al., 2020 a bit more explicitly explain why it isn't SecAgg compatible.

## Minor Requests

* In Section 1, you mention that you study cross-device FL, and give the objective function (1). This is slightly pedantic, but that objective is relevant to both cross-device and cross-silo FL. My interpretation is that the focus on cross-device is because you are concerned with the amount of client-to-server communication. This would be useful to clarify (maybe even introducing briefly what cross-device FL is).
*  Adding a reference to systems-level discussion of FL to back up the discussion on communication bottlenecks in Section 1.1 would be appreciated. Related to the above, it would be useful to clarify that this is generally a cross-device FL issue, not cross-silo.
*  Please add citations related to the SecAgg protocol, and define explicitly what SecAgg compatiblity means in this context (ie. that the server can only take sums of client values) for readers who may not know.
*  There is a missing close parenthesis in equation 16.
*  The authors have a few instances (eg. on page 10) where footnotes are appended to numbers, making it look like an exponent. If possible consider slightly changing these for readability.
*  The citation formatting is slightly inconsistent (eg. whether or not to capitalize all words in a conference title). Please standardize them.
*  At least one of the papers being cited (QSGD) exists in published form, but the arXiv version is cited. Please cite the published form when possible.
*  Since federated learning is a concept (like machine learning) it should not be capitalized unless in a title.
*  There is a reference to (unchanged) before the Shakespeare dataset (Section 5.1.2). Is this simply a reference to the fact that you do not change the number of examples held by clients? This might be useful to clarify.

**Strengths And Weaknesses:**

The paper is generally well-written, and easy to follow. The analysis of the client sampling mechanism is elegant and encourages the reader to follow along in the reduction of the problem to something tractable. While the convergence analysis requires a bit more machinery and a variety of technical constants, the authors do a good job of putting forward intuitive interpretations of their results. Finally, the experimental section is well-documented and, as best as I can tell, fully reproducible (save for the partitions of EMNIST they use, as I will discuss below).

Overall, I found this paper compelling from a message standpoint. The authors clearly outline realistic, practical methods for reducing total communication costs in FL (though there is some nuance here which I discuss below) and do a great job of transitioning these goals into interesting theoretical questions. The authors clearly support most of the claims made at the beginning of the work (save for one notable exception which I detail below). There are sections which I found more interesting and compelling than others. As such, I have separated my review according to the 3 major sections of the paper (2, 3, and 5).

### Section 2 - Sampling Schemes and Optimality

I will preface this discussion with the fact that I think Section 2 is well-written, concise, and well-motivated. Moreover, I enjoy that while the work is primarily a theoretical analysis, the authors do a good job of trying to design algorithms with SecAgg and system practicality in mind.

The primary weakness of this portion of the work is that most of the optimality results have been derived by previous works, namely [1, 2]. While these works were not directly about FL, the idea of minimizing variance of some linear combination subject to a sparsity constraint is more general, and the exact same sampling scheme, with proofs of optimality, has been shown previously in [2]. Optimal sampling schemes for the reverse problem (minimize sparsity subject to a variance constraint) are similar, and have been derived in [1]. The observation that independent sampling schemes are optimal among all set sampling schemes is interesting and not present in those works however. It is also worth mentioning that Algorithm 2 is a minor modification of Algorithm 3 in [1]. While I really like the writing, clarity, and pertinence to FL of Section 2, I believe that its current positioning by the authors does a disservice to the paper.

That is, instead of trying to argue that the authors provide a "novel adaptive partial participation strategy" (section 1.2) and that they "obtain an approximation to our optimal sampling strategy which...[fulfills] the privacy requirements of FL" (section 1.2), I think that the core interesting observation of the paper is that sparsification schemes/algorithms previously designed in the literature have renewed (and perhaps increased) relevance in the context of FL, especially methods that are compatible with SecAgg.

One last note on Section 2 - I think it does a good job of discussing some aspects of FL (eg. why compatibility with SecAgg makes certain problems much more challenging) it makes some assumptions about realistic systems that may or may not hold in practice. In particular, the authors explicitly tackle "total amount of communication per round" as the metric to minimize (see Section 2.2, "Extra Communication Costs"). However in practice, engaging in multiple synchronous rounds of communication (as in Algorithm 2) can also be a bottleneck [3]. I do not think this is a problem that needs to be solved by the authors. Rather, I like the focus on total communication cost, and would like to see some acknowledgment that if this is not the metric you want, then you may not want to use Algorithm 2.

### Section 3 - Convergence Analysis

Section 3 and the convergence analysis therein is the strongest part of this work. In particular, the authors relax a number of strong assumptions that many other works make (such as a uniformly bounded dissimilarity, instead of a bounded dissimilarity on average, see Assumption 3.5). I also greatly appreciated the less technical "Interpretation" sections, which capture useful historical comparisons to great effect (though I would request that the authors use equations to spell out what convergence rates in other works are, just to make the comparisons easier to parse).

One slight weakness of the section is the relative opacity of the results on FedAvg. While some of the edge cases are more interpretable, the overall results involve a number of interrelated terms that make it difficult to reason about which parts dominate the convergence result. While this is to some extent unavoidable, an expanded discussion of the FedAvg convergence results could be useful here. Similarly, a brief sketch of the how the proofs go (in particular, how does the sampling scheme alter existing convergence theory of FedAvg) would go a long way.

### Section 5 - Experimental Results

Section 5 is thorough in many ways, and conveys to good effect the utility of the method, but stops short in a few important ways. I will preface this with saying that given the theoretical focus of the work, I do not think that the authors need to provide a comprehensive comparison of their method to many other proposed sampling schemes, especially in light of the focus on SecAgg-compatibility.

That being said, I think there are 2 notable weaknesses of this section. First, as best as I can tell the authors do not describe how they partition the FEMNIST dataset into Datasets 1, 2, and 3. While I appreciate why they do this split (in order to simulate more interesting environments for this kind of approach), they simply say that they go about "removing some images from some clients." I believe that the mechanism by which this was done should be more explicit.

Second, I would have liked to see experiments with different values of $m$. In both the FEMNST and Shakespeare settings, $m$ is set to be relatively small ($m  \in \{2, 3\}$) or else is set to the number of participants $n$. Understanding the transition between small and large $m$ would be useful. For example, one natural type of question is what kind of relation between $m, n$ is sufficient to obtain nearly identical results to $m = n$ (eg. constant factor, square root, logarithmic, etc.). This kind of investigation would also benefit from other values of $n$.

### Review References

[1] Wangni et al. "Gradient sparsification for communication-efficient
distributed optimization." NeurIPS 2018.

[2] Wang et al. "Atomo: Communication-efficient learning via atomic sparsification." NeurIPS 2018.

[3] Huba et al. "PAPAYA: Practical, Private, and Scalable Federated Learning." MLSys 2022.

---

> ### Author Response · Authors · 2022-07-05
> **Thank you for the very insightful review**
>
> Thank you for this insightful and helpful review. We appreciate all the tremendous effort that the reviewer spent on the paper.
>
> We hope that we have sufficiently addressed all the critical and recommended changes and that the reviewer could champion our paper.
>
> All the changes are in the revised manuscript highlighted in blue.

---

### Review · Reviewer_hTeJ · 2022-06-16

**Summary Of Contributions:**

In this paper, the authors propose a new statistical client sampling approach to tackle the communication bottleneck issue of Federated Learning (FL) which relies on the updates of "more informative" clients in each communication round.
The proposed “optimal” sampling strategy is formulated based on minimizing the variance of the estimated gradient over (independent sampling) clients which then can be solved by a closed-form formula using only the norms of the updates.
Theoretical analyses and convergence guarantees are provided for the proposed strategy with the Distributed SGD and FedAvg algorithms.
Experimentally, the authors show that their strategy performs betters in terms of communication complexity than the uniform client sampling strategies.


**Broader Impact Concerns:**

N/A.

**Requested Changes:**

- I would recommend the authors clearly state the novelty and the differences of their proposed problem formulation from other works on importance sampling. Furthermore, to make the experiment results to be more convincing, the authors should compare their work with other client sampling baselines.

**Strengths And Weaknesses:**

Strengths:

An efficient client sampling scheme with theoretically motivated has been introduced to reduce communication costs, one of the primary challenges in FL settings.
The convergence analysis for the proposed approach is provided with some reasonable assumptions.
The proposed algorithms are compatible and can be adapted to existing local update methods (DSGD, FedAvg) which keeps the nature of secure aggregation and privacy-preserving in FL.
Numerical results show the proposed sampling scheme communication gains compared to uniform sampling approaches in a wide range of datasets.
The paper is well written and the concepts are explained in technical detail.

Weaknesses:
- Although experiments show that the results of the paper outperform uniform sampling, there is no comparison with existing research [1] and [2].
- Authors should make a comparison with existing research which are capable of adaptively finding w_i such as [3] and [4]. If we consider client selection objective function as a problem of looking for clients obtaining high impact on accuracy, [3] and [4] can point out clients who have significant contributions to the global model. Furthermore, [4] show that it not only achieves good performance in terms of accuracy but also reduces the number of communication between servers and clients.
- Experiments show that the number of selected clients for training has an impact on the accuracy of the global model. It is expected that authors should make more experiments for illustrating the effect of m on communication and processing costs in terms of bit at the central server.
- With non-i.i.d. data, some clients may hold data that other clients do not have. It is reasonable that clients holding a minor proportion of datasets still play an important role in the global model. Therefore, uniform sampling may have higher performance than the proposed method in this scenario.
- I do not feel this paper made sufficient novel contributions as the sampling scheme is designed mainly based on Lemma 2.1, an extension of the existing optimal sampling scheme that has been studied extensively in other literature.
S. Horvath and P. Richtarik, “Nonconvex variance reduced optimization with arbitrary sampling,” in International Conference on Machine Learning, 2019, pp. 2781–2789 (Lemma 1)
- The authors only compare their proposed method with uniform sampling and full participation, and "chose not to compare with other client sampling methods, as such comparisons would be unfair". It's quite not convincing me as many efficient client sampling approaches for improving FL in the existing literature.

Luo, B., Xiao, W., Wang, S., Huang, J., & Tassiulas, L. (2021). Tackling System and Statistical Heterogeneity for Federated Learning with Adaptive Client Sampling. ArXiv, abs/2112.11256.
H. T. Nguyen, V. Sehwag, S. Hosseinalipour, C. G. Brinton, M. Chiang, and H. V. Poor, “Fast-convergent federated learning,” IEEE Journal on Selected Areas in Communications, vol. 39, no. 1, pp. 201–218, 2021.
H. Yang, M. Fang, and J. Liu, “Achieving linear speedup with partial worker participation in non-iid federated learning,” arXiv preprint, arXiv:2101.11203, 2021.
Diverse Client Selection for Federated Learning: Submodularity and Convergence Analysis, ICML2021
Peilin Zhao and Tong Zhang. Stochastic optimization with importance sampling for regularized loss minimization. In International Conference on Machine Learning, pp. 1–9, 2015

---

> ### Author Response · Authors · 2022-07-05
> **Response to the mentioned weaknesses**
>
> Below, we provide our response to the reviewer's concerns that we believe can be sufficiently addressed with an extra explanation as discussed next.
>
> (in order of reviewer's comments)
>
> 1. Please note that the full participation baseline is the strongest possible baseline in theory regarding the communication rounds, and our proposed method matches its performance. This already empirically shows the optimality of our approach. Therefore, the room for improvement is minimal. More importantly, both works mentioned by the reviewer are incompatible with the stateless clients and secure aggregation and thus are not comparable with our method. We have cited them in the related work section.
> 2. Please note that our results are consistent with the work of Yang et al. (2021). We also obtain linear speed up when we increase the number of workers. We have added discussions and a citation to them in section 5.2. The work of Balakrishnan et al. (2021) cites our work as prior work on sampling, so we omit the comparison. However, we advise the reviewer not to check for the citation at this stage, as this would violate the double-blind reviewing policy implemented by TMLR.
> 3. We have added extra experiments with different values of $m$ in the updated manuscript. The extra results are consistent with our theory (i.e., larger $m$ leads to better communication complexity and faster convergence).
> 4. We kindly disagree with the statement that uniform sampling should lead to better performance than our proposed method. Our proposed sampling strategy should be able to filter such unimportant updates as we sample based on the "importance" (norm) of the update. Our experiments also support this, as we consistently outperform the uniform sampling baseline by a large margin. We are happy to conclude extra experiments to provide further evidence if the reviewer has a concrete example in mind.
> 5. We do not claim that the used sampling scheme is new. In fact, we build directly upon the prior work of Horvath & Richtarik (2019). The main contribution of our paper lies in properly applying this sampling procedure to the FL framework, leading to a superior method for client sampling in FL while not compromising privacy: we note that our proposed sampling is the first optimal importance sampling method that is compatible with two core privacy requirements of FL: secure aggregation and statelessness of clients. We would also like to kindly remind the reviewer that the "novelty of the studied method is not a necessary criterion for acceptance" in TMLR.
> 6. Please note that the full participation baseline is the strongest possible baseline in theory regarding the communication rounds, and our proposed method matches its performance. This already empirically shows the optimality of our approach. Therefore, the room for improvement is minimal. For the "unfair comparison," we refer to the fact that none of the methods we are aware of is compatible with both secure aggregation and statelessness of clients.

---

### Review · Reviewer_k75y · 2022-06-20

**Summary Of Contributions:**

This paper studies how to reduce the communication cost in federated learning by allowing the server to communicate with only "important" clients. Based on this insight, the authors proposed a new client sampling strategy which can minimizes the gradient variance, and is compatible with stateless clients and MPC protocols. Convergence analysis and experimental results further validate the effectiveness of the proposed method.

**Requested Changes:**

1. Change the title and related discussions in the paper. The method is optimal in reducing variance but it is not the optimal among all client sampling strategies.
2. $R$ denotes the number of local steps in FedAvg. But $R_i$ is defined as $f_i(x^*)-f_i^*$. It would be better to change one of these notation.
3. The experiments should be conducted in an environment where all clients perform the same number of local steps.
4. Revise the introduction to reflect the fact that the idea of using client sampling to reduce communication is not new.
5. More experiments on each dataset to show how the performance changes with $m$.

**Strengths And Weaknesses:**

**Strengths**
1. The authors pointed out all the previous client sampling schemes are not compatible with stateless clients and MPC protocols. They also come up with some strategies to overcome this limitation. This part is very novel and can be interest of future researchers.
2. The proposed client sampling strategy has convergence guarantee. The theory shows that the proposed method can have better convergence guarantee than vanilla FedAvg.

**Weaknesses**
1. The title "optimal client sampling for federated learning" is very controversial. I understand that the proposed sampling strategy is optimal because it minimizes the gradient variance.So we can say it is an optimal strategy to minimize the gradient variance. However, there can be a lot of different client sampling strategies for federated learning. Instead of minimizing the gradient variance, one can also propose some other strategies to minimize other metrics. It does not make sense to claim this particular method with minimal gradient variance is optimal among all client sampling strategies.
2. In theory, the improvements from the proposed client sampling scheme seem to be marginal. Specifically, it just changes the constant factor of the non-dominant term in the error upper bound. When the number of communication rounds are sufficiently large, the proposed method should have the same convergence rate as the vanilla FedAvg.
3. I do not think the experimental results are convincing. The theory assumes all clients have the same number of local steps. But the experiments do not. In particular, the authors let all clients perform 1 epoch of local training but clients have drastically different amount of local data. As a result, the number of local steps are different across clients. It is likely that the clients with more local steps would have larger norms of local updates. So they would get more chances to be selected by the server. The proposed sampling method just prefers those clients with more data. I guess when all clients have the same number of local steps, maybe the benefit of the proposed method can be very incremental? since all clients may have the same level of local-update norms.
4. Although the proposed method can reduce the communication bandwidth, it still requires all available clients to communicate with the server. So the communication latency or the delay in establish communication links between server and clients are not saved. In the settings where the comm. latency is quite high, the proposed method may not be effective in reducing the real communication time.
5. In the introduction, the authors first overview two methods of reducing communication cost in federated learning (allowing local updates and compression). Then, they suddenly jumped into contribution and said "we propose a new approach to address the communication bandwidth issue". This sounds like the idea of client sampling is first proposed in this paper. But it does not. It has appeared in the original FedAvg paper and a lot of subsequent client sampling papers. The idea "careful selection of clients can lead to better communication complexity" also appears in many previous literature, as the authors discussed in the related works section.
6. In the experiments, this is just one specific value of $m$ for each dataset. I feel it is necessary to understand how the performance of the proposed method changes along with $m$.

---

> ### Author Response · Authors · 2022-07-05
> **Response to reviewer's concerns**
>
> Below please find our response to the raised concerns:
>
> 1. Thank you for raising this comment, but we would kindly disagree with the statement that our title is controversial. Let us elaborate on this. Optimizing the left-hand side of (5) under the communication budget, i.e., $\sum p_i \leq b$, leads to the optimal sampling among all sampling strategies. Other works, such as Cho et al. (2020), use different metrics, e.g., client loss, which are only proxies for minimizing (5). In our work, we minimize its upper bound that holds as equality for the case $b=1$ or if the target sampling is independent across clients, which is a reasonable assumption considering the privacy requirements of FL. We kindly refer to the remarks in Section 2.1 and Section 4.1 for detailed discussion.
> 2. We kindly disagree with this statement. As discussed below, for each convergence theorem, the nature of our improvement is equivalent to the increasing cohort size despite only a few clients communicating their full updates to the master node, which also improves the statistical term, which becomes dominant for the sufficiently large number of communications rounds. Our experimental results also confirm this.
> 3. The nature of our experiments and the analysis is consistent with the literature, e.g., the first FL paper of McMahan et al. (2015) also contains this discrepancy. We realize that such consideration leads to objective inconsistency. We are happy to add the discussion to state this clearly. We will refer to the work of Wangni et al. 2020 (https://proceedings.neurips.cc/paper/2020/file/564127c03caab942e503ee6f810f54fd-Paper.pdf) and Horvath et al. 2022 (https://arxiv.org/abs/2204.13169). Furthermore, while it might be a case that clients that run more local steps have larger norms in some scenarios, this is not always true. For instance, for the clients for which the global model performs already well, i.e., “solve” their local problem, the update will always be of small magnitude regardless of the number of local steps. Practically, our method will be able to filter such updates out. We will run experiments with a balanced (each client has the same amount of data) CIFAR100 dataset and also analyze the norm of the updates to provide better evidence for the practical performance of our method.
> 4. Thank you for raising this issue. We have added a comment about this in the conclusion section. Indeed, if establishing communication links is the main bottleneck, our method will not lead to much of an improvement. However, we note that in the case where the actual communication is the bottleneck in the system, our method leads to significant gains, as shown both in theory and practice.
> 5. We do not claim to propose the first importance sampling strategy. We argue that the proposed method is the first importance client sampling strategy that is compatible with two core privacy requirements of FL: secure aggregation and statelessness of clients. We have updated the wording in the introduction to avoid this confusion.
> 6. We have added extra experiments with different values of $m$ in the updated manuscript. The extra results are consistent with our theory (i.e., larger m leads to better communication complexity and faster convergence).

---

### Author Response · Authors · 2022-07-05
**General response to all reviewers**

We want to thank all the reviewers for their valuable time and feedback. We apologize for the late reply and hope to still engage in conversation.

We particularly appreciate that:
 -  **Reviewer k75y** identified important strengths mentioned in our manuscript, namely that our sampling scheme is the first compatible with stateless clients and MPC protocols and that we provide a strong convergence guarantee showing better rates than vanilla methods, i.e., without our importance sampling strategy.
-   **Reviewer hTeJ** also identified that the considered sampling solves a significant problem, is efficient, and comes with strong theoretical guarantees reducing the communication cost of FL. The reviewer also recognizes the compatibility with existing popular optimization methods and privacy-enhancing techniques as an important feature of the presented approach.
- **Reviewer w6Qr** finds the paper overall compelling with significant and relevant contributions.
- two reviewers also acknowledge that the paper is well written and easy to follow.

We address the reviewers' concerns in separate comments below respective reviews. Overall, we believe the raised issues are well addressed, and we hope the reviewers will be pleased with our response. As requested by the reviewers, we have updated the manuscript, where all the changes are highlighted in blue.

---

### Decision · Action_Editors · 2022-08-03

**Recommendation:** Accept with minor revision

**Comment:**

Based on the reviews, the author response/revision, the subsequent discussion and the reviewers' recommendations, my decision recommendation for this paper is **Accept with minor revision**.

The rationale behind this decision is that it that the only point that remains unaddressed in the revised version is the lack of experiments where all clients perform the same number of local updates. While the paper would be stronger with this additional experiment, I do not think this warrants rejection because, even if the proposed approach turns out to be less effective in this setup, the contribution remains valuable as per TMLR's criteria.

For the paper to be accepted, **the authors must add the experiment promised in their response to reviewer k75y**:
> We will run experiments with a balanced (each client has the same amount of data) CIFAR100 dataset and also analyze the norm of the updates to provide better evidence for the practical performance of our method.